# The Nexus of Inflammation-Induced Epithelial-Mesenchymal Transition and Lung Cancer Progression: A Roadmap to Pentacyclic Triterpenoid-Based Therapies

**DOI:** 10.3390/ijms242417325

**Published:** 2023-12-10

**Authors:** Kirill V. Odarenko, Marina A. Zenkova, Andrey V. Markov

**Affiliations:** 1Institute of Chemical Biology and Fundamental Medicine, Siberian Branch of the Russian Academy of Sciences, 630090 Novosibirsk, Russia; k.odarenko@yandex.ru (K.V.O.); marzen@niboch.nsc.ru (M.A.Z.); 2Faculty of Natural Sciences, Novosibirsk State University, 630090 Novosibirsk, Russia

**Keywords:** inflammation, pulmonary malignancy, epithelial-to-mesenchymal transition, natural products, aggressiveness, tumor stem cells, mechanism of action

## Abstract

Lung cancer is the leading cause of cancer-related death worldwide. Its high mortality is partly due to chronic inflammation that accompanies the disease and stimulates cancer progression. In this review, we analyzed recent studies and highlighted the role of the epithelial–mesenchymal transition (EMT) as a link between inflammation and lung cancer. In the inflammatory tumor microenvironment (iTME), fibroblasts, macrophages, granulocytes, and lymphocytes produce inflammatory mediators, some of which can induce EMT. This leads to increased invasiveness of tumor cells and self-renewal of cancer stem cells (CSCs), which are associated with metastasis and tumor recurrence, respectively. Based on published data, we propose that inflammation-induced EMT may be a potential therapeutic target for the treatment of lung cancer. This prospect is partially realized in the development of EMT inhibitors based on pentacyclic triterpenoids (PTs), described in the second part of our study. PTs reduce the metastatic potential and stemness of tumor cells, making PTs promising candidates for lung cancer therapy. We emphasize that the high diversity of molecular mechanisms underlying inflammation-induced EMT far exceeds those that have been implicated in drug development. Therefore, analysis of information on the relationship between the iTME and EMT is of great interest and may provide ideas for novel treatment approaches for lung cancer.

## 1. Introduction

Lung cancer is one of the most commonly diagnosed cancers and the leading cause of cancer-related deaths worldwide, with an estimated 2,207,000 new cases and 1,796,000 new deaths in 2020 [1]. In Russia, lung cancer ranks second in overall incidence and first in males, with a 5-year survival rate of 15–20% [2]. Despite significant advances in diagnostic and surgical techniques, as well as targeted drug development, the mortality of patients with lung cancer is high and steadily increasing. One of the major reasons for this failure is the lack of therapeutics that are able to effectively control the crosstalk between cancer and the tumor microenvironment (TME), leading to the enhancement of tumor aggressiveness and metastasis.

Numerous studies have shown that stromal and immune cells surrounding tumor tissue are capable of producing various growth factors and cytokines that induce the epithelial–mesenchymal transition (EMT) in cancer cells [3]. Malignant cells undergoing EMT are characterized by the disruption of adherens junctions, the cadherin switch from E-cadherin to N-cadherin, and cytoskeletal reorganization, leading to the acquisition of a motile and invasive phenotype [3]. Moreover, a range of evidence has been collected that suggests that the EMT of lung cancer cells mediates not only the enhancement of their metastatic potential but also their evasion of the immune system and the development of drug resistance [4], clearly indicating an aggravating role of EMT in lung cancer pathology. Indeed, a significant positive correlation between low E-cadherin/high vimentin expression (an important EMT feature) and worse overall survival of lung cancer patients has been reported [5].

It is known that inflammatory airway injury induced by various stressors, including tobacco smoke, industrial dust, allergens, and pulmonary infections, not only creates a milieu conducive to lung carcinogenesis and is associated with a high risk of lung cancer [6] but also promotes the EMT in the lung [7]. For example, an active EMT has been identified in the airway epithelium of smokers and patients with chronic obstructive pulmonary disease (COPD) [8] as well as in severe COVID-19 patients [9]. Because of the established links between chronic lung inflammation, the EMT, and lung cancer aggressiveness, therapeutic modulation of an inflammation-driven EMT can be considered a promising approach in the treatment of pulmonary malignancies.

One of the most important sources of bioactive compounds capable of suppressing tumor progression is pentacyclic triterpenoids (PTs), multitarget plant metabolites that effectively suppress the proliferative, migratory, and invasive capacities of tumor cells [10]. Moreover, these molecules have been shown to significantly block the EMT of tumor cells of different origins driven by various EMT inducers, including pro-inflammatory mediators [11,12,13]. Given this fact, as well as the demonstrated ability of PTs to ameliorate pulmonary inflammation in vivo [14], these compounds can be considered a promising platform for the development of novel blockers of inflammatory-associated EMT in lung cancer.

In this review, we summarize recent insights into the relationship between the inflammatory background and EMT-driven enhancement of the malignant traits in lung cancer cells and shed light on the possibility of using PTs as potential inhibitors of this interconnection. In view of the pandemic spread of SARS-CoV-2 and COVID-19-associated severe pulmonary pathologies, the knowledge gained in this review will be useful for both lung oncology and medicinal chemistry researchers.

## 2. Inflammatory-Driven EMT of Lung Cancer Cells: Key Players and Regulators

### 2.1. EMT-Associated Changes in Cancer Cells

The epithelial–mesenchymal transition (EMT) is a process of epithelial cell phenotype switching that plays a crucial role in a plethora of normal physiological processes, including embryonic development, wound healing, and tissue regeneration, as well as pathological processes such as fibrosis, neoplastic transformation, and cancer metastasis [15].

Various stimuli from TME, such as growth factors [16], hypoxia [17], low pH [18], or changes in the extracellular matrix (ECM) [19], upregulate EMT-associated transcription factors (EMT-TFs) of the Snail (Snail, Slug), ZEB (ZEB1, ZEB2), and Twist (Twist1, Twist2) families, which in turn shift the balance of EMT-associated cell markers toward mesenchymal ones (N-cadherin, vimentin, fibronectin, etc.) [20] (Figure 1).

The main feature of the EMT, an increase in the migratory and invasive abilities of tumor cells, results from a complex process of transdifferentiation involving many cellular components. First, E-cadherin, an epithelial cell–cell adhesion protein, is replaced by N-cadherin, which leads to the formation of relatively weak adhesion junctions [21]. This process leads to the release of β-catenin from the cadherin complex, followed by its translocation to the nucleus (Figure 1). There, β-catenin triggers the expression of EMT-TFs through the engagement of LEF/TCF transcription factors, thus forming a positive feedback loop to maintain the mesenchymal phenotype [21]. In addition to changes in surface markers, cytoskeletal rearrangements occur during EMT and enhance cell motility. The replacement of cytokeratin filaments with vimentin filaments increases the mechanical strength of the cytoskeleton by promoting microtubule polarization, stress fiber formation, and focal adhesion stabilization [22].

Alterations in the cytoskeleton and cell–cell adhesion together lead to a change in cell morphology: the typical apical-basal polarity of epithelial cells is replaced by a front-rear polarity of rapidly migrating mesenchymal cells [23]. In addition to changes in the cellular architecture, the EMT is accompanied by extensive ECM rearrangements, which in turn provide molecular support to cells to complete their transition. For example, fibronectin fibers secreted by mesenchymal-like cells bind EMT-inducing growth factors and facilitate their delivery to receptors [24]. Collagen I stabilizes Snail by activating the discoidin domain receptor 2 (DDR2) and the downstream Src/ERK2 signaling pathway [25]. Simultaneous to the formation of the new ECM composed of fibronectin and collagen I, cells undergoing an EMT cleave collagen IV and laminin of the basal lamina by secreting proteolytic enzymes such as matrix metalloproteinases 2 and 9 (MMP-2 and MMP-9, respectively) and thus invade surrounding tissues [26] (Figure 1).

Altogether, the EMT leads to the disruption of cell–cell interactions, loss of apical-basal cell polarity, cytoskeletal rearrangements, and the acquisition of a motile invasive phenotype by tumor cells. These changes ultimately cause cells to migrate from the epithelium, enter the circulation, and disseminate to distant sites, where they can undergo a reverse transition, giving rise to micro- and macrometastases [27].

### 2.2. Inflammatory TME and Its Role in EMT Induction in Lung Cancer Cells

Inflammation is known to play a crucial role in the initiation and maintenance of tumor growth by providing pro-tumorigenic components to the TME [28]. Given that patients with diseases such as COPD and pulmonary fibrosis that progress to chronic inflammatory conditions have a higher risk of developing lung cancer [29,30], pulmonary inflammation can be considered an important participant in the malignant transformation of lung tissue. Moreover, pro-inflammatory cytokines produced in the microenvironment of established lung tumors have a pronounced pro-metastatic effect, including the stimulation of EMT [31]. To better understand the key aspects of inflammation-driven lung cancer progression, in this section, we focus on the cellular landscape that forms the inflammatory TME (iTME) and its impact on EMT induction in lung cancer cells (Figure 2).

#### 2.2.1. Cancer-Associated Fibroblasts

There are two complementary pathways contributing to the formation of iTME, including the intrinsic pathway, when genetical changes in neoplastic cells drive the overproduction of pro-inflammatory mediators, and the extrinsic pathway, when iTME is induced by persistent infection, autoimmune disorders, or chronic exposure to an irritant [32]. Along with tumor cells, cancer-associated fibroblasts (CAFs), a heterogenous cell population originating from normal fibroblasts, adipocytes, epithelial and endothelial cells, bone marrow-derived mesenchymal stem cells, and some other cell types, play a key role in tumor stroma formation [33]. CAFs are able to produce ECM components and various growth factors, creating favorable conditions for the proliferation and survival of tumor cells, as well as cancer invasion and neoangiogenesis [34] (Figure 3).

To date, a large number of data have been accumulated confirming the ability of CAFs to induce the EMT of lung cancer cells [35,36,37,38,39,40,41,42]. TGF-β secreted by CAFs plays an important role in this process: co-incubation of lung adenocarcinoma A549 and NCI-H358 cells with CAF-derived conditioned medium and the selective TGF-β inhibitor SB431542 significantly suppressed CAFs-induced EMT progression [35]. In addition to TGF-β, the hepatocyte growth factor (HGF) and insulin-like growth factor 1 (IGF-1) secreted by CAFs participate in the control of EMT initiation in lung cancer cells by synergistically inducing ANXA2 expression and phosphorylation in tumor cells with subsequent cadherin switching and stimulation of vimentin expression, cell migration, and invasion [36].

In addition to growth factors, CAFs secrete a number of pro-inflammatory proteins with pronounced EMT-stimulating activity against lung cancer cells [37,38]. Shintani et al. found that the conditioned medium of CAFs is enriched in interleukin 6 (IL6), which induces the development of the EMT phenotype of non-small cell lung cancer (NSCLC) cells and mediates the development of cell resistance to cisplatin [37]. In an independent study by scientists from Tianjin Medical University General Hospital, the EMT-stimulating effect of IL6 produced by CAFs was shown to be based on the activation of the JAK2/STAT3 signaling pathway: it was found that the pretreatment of lung cancer cells with JAK2- and STAT3-specific inhibitors, AG490 and Stattic, respectively, significantly reversed the effect of the CAF-conditioned medium on the regulation of EMT- and metastasis-related genes and cell motility [38]. In addition to IL6, the promoting effect of CAFs on EMT programming in lung cancer cells is also mediated by the secretion of CXCL12 and high mobility group box 1 (HMGB1), which activate the CXCR4/β-catenin/PPARδ [39] and NFκB [40] signaling pathways, respectively.

A number of recent studies have demonstrated that CAFs are able to deliver EMT-stimulating signals to lung cancer cells via exosomes [41,42]. You et al. reported that the exosome-mediated secretion of Snail by CAFs resulted in the suppression of E-cadherin expression and upregulation of vimentin, accompanied by increased motility and invasiveness of A549 and H1299 cells [41]. Yang et al. found that CAFs isolated from patients with lung adenocarcinoma secreted exosomes containing miR-210, which significantly enhanced the migration and invasion abilities of NSCLC cells via the activation of the PTEN/PI3K/AKT signaling pathway [42]. Since both Snail and miR-210 are known to be associated not only with EMT but also with immune regulation [43,44], these molecules can also be considered important components of iTME that enhance the malignancy of lung cancer cells.

#### 2.2.2. Immune Cells

Immune cells of both innate and adaptive immunity are another cellular component of the tumor stroma that has a multidirectional influence on cancer progression (Figure 2). Chae et al. analyzed the immune landscape of tissue samples from patients with lung adenocarcinoma and lung squamous cell carcinoma according to the “epithelial” or “mesenchymal” tumor phenotype, which differed in the expression of 16 EMT-associated genes [45]. The tissues with a “mesenchymal” phenotype, i.e., containing cells that have undergone EMT activation, show a lower infiltration of CD4+/CD8+ T cells and dendritic cells, which play a key role in immune surveillance against cancer, as well as increased infiltration of B cells and regulatory T cells (Tregs). The secretion of immunosuppressive cytokines such as IL-10, IL-4, and TGF-β by B cells and Tregs, in addition to those produced by tumor cells, allows them to escape immune surveillance and leads to subsequent tumor progression and metastasis [45]. These data agree well with the results of in vitro experiments using a co-culture system consisting of peripheral blood mononuclear cells from healthy donors with A549 cells treated with a combination of inflammation-related cytokines (TGF-β, TNF-α, and interferon γ (IFN-γ)) or the supernatant derived from a mixed lymphocyte reaction. A549 cells that have undergone EMT activation under these stimuli have been shown to acquire immunomodulatory properties, such as the ability to increase or decrease the proliferation of B cells and T cells, respectively, which is in good agreement with the levels of these cell types in “mesenchymal” type lung tumors [45,46].

##### Granulocytes

Some of the aforementioned immunomodulatory molecules exhibit pleiotropic effects, and in addition to their immunomodulatory activity, they directly or indirectly induce EMT in tumor cells (Figure 2). TGF-β is an anti-inflammatory cytokine with this mode of action. On the one hand, it induces the formation of a tumor-associated N2 neutrophil phenotype, which plays a role in supporting tumor growth and suppressing the antitumor immune response [47]. On the other hand, TGF-β triggers the EMT in lung cancer cells [48]. Interactions between these cell types have been demonstrated in co-culture experiments, revealing that N2 neutrophils contribute to the induction of EMT in A549 cells through the activation of the TGF-β/Smad signaling pathway [49] (Figure 4A). This signaling cascade is also activated in human bronchial epithelial BEAS-2B cells after incubation with primary human eosinophils or human eosinophilic leukemia Eol-1 cells [50,51]. Furthermore, cysteinyl leukotrienes (CysLTs), pro-inflammatory mediators contributing to the pathogenesis of chronic asthma [52], also play a crucial role in eosinophil-induced EMT. Treatment of BEAS-2B cells co-cultured with eosinophils using a selective CysLT inhibitor significantly suppressed TGF-β production, resulting in the blockade of the TGF-β/Smad3 signaling axis and, consequently, the inhibition of the expression of EMT markers and EMT-mediated changes in the morphology of BEAS-2B cells [51] (Figure 4A).

##### Tumor-Associated Macrophages

In addition to the previously discussed neutrophils and eosinophils, tumor-associated macrophages (TAMs) have been implicated in the formation of iTME. Macrophages are conventionally associated with tissue development and healing due to their capacity to remodel the extracellular matrix (ECM), generate growth and angiogenesis factors, and engulf apoptotic cells [53]. However, within the tumor microenvironment, these macrophage properties take on a pathological role, supporting cancer invasion and metastasis [54]. Several cytokines secreted by macrophages, including TNF-α, IL-6, and IL-8, have the potential to induce EMT in tumor cells (Figure 4B), thereby promoting metastasis [55,56].

Using a co-culture of human lung adenocarcinoma A549 and H1299 cells with THP-1-derived macrophages, Dehai et al. demonstrated a bidirectional interaction between neoplastic epithelial cells and macrophages [57]. Tumor cells stimulate macrophage differentiation toward the M2 phenotype, which is involved in immunosuppression, angiogenesis, and cancer invasion. Macrophages, in turn, enhance the invasive ability of tumor cells and induce an EMT, as evidenced by the switch in cadherin abundance (from E-cadherin to N-cadherin). The co-culture of THP-1-derived macrophages with A549 or H1299 cells resulted in the upregulation of IL-6 in co-cultured cell lines, and IL-6 was also found to induce the EMT in A549 and H1299 cells (Figure 4B). However, the blockade of the EMT by neutralizing anti-IL-6 antibody was less pronounced in the co-culture system compared to that observed in cells treated with recombinant IL-6, suggesting that some other molecules in the TME are involved in the EMT [57]. These results are in good agreement with a recent study by Qin et al., which showed that the treatment of co-cultured A549 and THP-1 cells with *E. coli* lipopolysaccharide (LPS) shifted the expression profile of EMT markers in tumor cells toward the mesenchymal state and increased the secretion of IL-6 and TGF-β by both cell types [58]. Furthermore, the IL-6-related JAK2/STAT3 pathway was shown to be involved in EMT induction in this experimental model, indirectly confirming the EMT-inducing effect of IL-6 in the co-culture system [58]. The EMT-inducing effect of IL-6 was also confirmed in the human bronchial epithelial BEAS-2B cell model, as cell treatment with IL-6 shifted the expression profile of EMT markers toward the mesenchymal state, increased the expression of EMT-TFs Snail, Twist, and ZEB1, and enhanced cell migration and invasion in BEAS-2B cells [58]. Chen et al. further demonstrated that another TME factor, monocyte chemoattractant protein 1 (MCP-1/CCL2), can enhance IL-6-mediated EMT by exerting a synergistic effect with IL-6 on STAT3 signaling, which causes the subsequent activation of Twist (Figure 4B). In addition, IL-6 and CCL2 have been shown to upregulate each other, forming positive feedback loops to maintain STAT3 signaling activity [56]. These findings are in good agreement with clinical data showing that in patients with NSCLC, serum IL-6 is negatively correlated with E-cadherin and positively correlated with the levels of N-cadherin and vimentin in tumor tissues and, consequently, with lymph node and distant metastasis [59].

##### Mast Cells

Mast cells are another iTME component involved in EMT induction and the stimulation of metastasis (Figure 4C). Mast cells communicate with lung epithelial cells by extracellular vesicles (EVs) transporting proteins, RNA, and DNA. Yin et al. have shown that HMC-1 mast cell-derived EVs carry TGF-β1 on their surface and are able to induce EMT in A549 cells through the Smad pathway. However, the mechanism of EVs-induced EMT may involve some other signaling pathways, as incubation with mast cell-derived EVs induced the phosphorylation of proteins involved in PI3K/Akt, HIF-1, NF-κB, and JAK/STAT pathways, as well as proteins regulating focal adhesion and tight junctions in A549 cells [60] (Figure 4C). The migration of lung adenocarcinoma A549 and H520 cells induced by mast cells can be further enhanced by chymase secreted by mast cells. Chymase cleaves E-cadherin, attenuates cell–cell adhesion in the lung epithelium, and increases MMP-9 expression in the cells [61].

#### 2.2.3. Other Pro-Inflammatory EMT-Inducing Mediators in iTME

In addition to the aforementioned TGF-β, IL-6, and IL-8, several other immunomodulatory proteins, such as TNF-α, IL-1β, IFN-γ, and IL-17, are capable of either directly inducing EMT or enhancing its induction by other cytokines [62,63]. For example, a mixture of TNF-α, IL-1β, and IFN-γ was shown to significantly enhance TGF-β-induced EMT induction in A549 cells via the upregulation of TGF-β receptor type I (TβRI) expression [64]. Recently, Li et al. conducted a thorough study on the mechanism of IL-1β-dependent EMT stimulation: they demonstrated a gradual change in the EMT-associated phenotype of A549 cells during long-term exposure to IL-1β, resulting in the retention of mesenchymal characteristics of tumor cells after removal of the inflammatory stimulus (a phenomenon called “EMT memory”) [62]. It was shown that the initial step of EMT induction, which is dependent on the EMT-TF Slug, is transformed into a Slug-independent chronic step during which E-cadherin expression is controlled by epigenetic modifications, including DNA methylation and the repressive histone modifications H3K27Me3 and H3K9Me2/3, resulting in a prolonged mesenchymal phenotype in the absence of IL-1β. EMT memory has also been demonstrated when A549 are chronically treated with TGF-β or TNF-α [62].

IL-17 is a pro-inflammatory cytokine mainly produced by activated CD4+ T cells (Th17), which has been shown to stimulate lung cancer metastasis [65]. The analysis of human lung adenocarcinoma tissues performed by Huang et al. showed that IL-17 expression was positively correlated with the expression of N-cadherin, vimentin, Snail1, Snail2, and Twist1 and negatively correlated with E-cadherin expression [63]. This may indicate that IL-17 is involved in EMT induction in lung adenocarcinoma cells, which was further confirmed by in vitro studies. IL-17 induces EMT in A549 and Lewis lung carcinoma (LCC) cells through STAT3 signaling [63]. In addition, the NF-κB pathway also regulates IL-17-mediated EMT through the upregulation of ZEB1, as blocking NF-κB activity with a selective inhibitor abrogates EMT-associated changes in cancer cells [66].

TGF-β and TNF-α also implicate NF-κB in the induction of EMT: mesenchymal-related A549 spheroid cultures induced by these inflammatory cytokines showed high levels of phosphorylation of IKK and its downstream targets IκBα and RelA, as well as overexpression of the NF-κB-dependent genes *IL-8* and *BIRC3*. Furthermore, the inhibition of NF-κB resulted in the suppression of Twist, Slug, and ZEB expression and reduced the invasive activity of A549 cells, as well as their ability to metastasize to the lung in a mouse xenograft model [67].

Prostaglandin E2 (PGE2), produced from arachidonic acid, is another EMT-stimulating factor in the iTME [68,69]. Cyclooxygenase 2 (COX-2), the rate-limiting enzyme in PGE2 synthesis, is constitutively expressed in many tumor types, including lung adenocarcinoma [68]. PGE2 was shown to suppress E-cadherin expression via the upregulation of ZEB-1 and Snail in COX-2-overexpressing A549 cells. Consistent with this, immunohistochemical staining of human lung adenocarcinoma sections showed that COX-2 was positively and negatively correlated with ZEB2 and E-cadherin expression, respectively [69]. Interestingly, PGE2 had an opposite effect on TGF-β-treated A549 cells, as PGE2 or the PGE2 receptor agonists EP4 and EP2 have been shown to suppress TGF-β-mediated actin reorganization, cell migration, and fibronectin and collagen I expression in A549 cells, although they did not affect the switch from E-cadherin to N-cadherin [70]. Thus, the effect of PGE2 on EMT seems to be determined by interactions with other EMT-associated pathways.

Interestingly, prostaglandin D2 (PGD2), also a COX metabolite like PGE2, induced EMT in A549 cells by stimulating TGF-β expression, and this cytokine was found to be a key master regulator of PGD2-driven EMT. The inhibition of TβRI by selective inhibitor SB431542 or short hairpin RNA-mediated gene silencing effectively abrogated the EMT-stimulating effect of PGD2 in A549 cells [71].

#### 2.2.4. ECM Components as EMT Regulators

Hyaluronan (HA), a polysaccharide component of the ECM, maintains tissue integrity due to its viscoelastic properties, but also regulates inflammation in a size-dependent manner: high-molecular-weight HA (~104 kDa) inhibits, whereas low-molecular-weight HA (~200 kDa) induces inflammation [72]. HA has been shown to be overproduced by tumor and stromal cells under inflammatory conditions [73].

TGF-β1 and IL-1β induce HA production by upregulating hyaluronan synthases (HAS) in lung adenocarcinoma cells [74]. The overexpression of HAS3 causes morphological changes, invasiveness, E-cadherin suppression, and vimentin induction in H358 cells, suggesting the role of HA as an autocrine factor in maintaining EMT [74]. When grown as 3D spheroids on HA-grafted chitosan membranes, A549 and H1299 cells exhibit higher levels of EMT markers and migration and invasion potential than their 2D counterparts [75]. Han et al. introduced mesenchymal stem cells (MSCs) into this system and demonstrated that HA provides structural support for their interaction with lung cancer cells. MSCs form a spheroid core and produce the EMT-inducing factors IL-10, TGF-β1, and CXCL12, which are thought to induce A549 cells from the spheroid margin to undergo EMT, as evidenced by molecular markers and increased motility in a zebrafish xenograft model [76]. HA interacts with hyaluronan mediated motility receptor (RHAMM) and CD44, and the inhibition of either of these proteins causes lung cancer cells to switch to an epithelial phenotype [77,78]. Furthermore, CD44+ primary lung adenocarcinoma cells show a clear EMT induction and have higher tumorigenic potential in nude mouse xenografts compared to CD44- cells [79]. Two molecular mechanisms of CD44-mediated EMT have been identified: (1) CD44 triggers the Wnt/β-catenin pathway, which causes FoxM1 to bind to the promoter and induce Twist [79]; (2) upon activation, CD44 colocalizes with epidermal growth factor receptor (EGFR) at the membrane and promotes the downstream activation of AKT and ERK pathways [80]. Interestingly, the disruption of CD44 expression inhibits EMT when A549 cells are treated with TGF-β1 alone, but not in combination with TNF-α, suggesting a close link between the HA/CD44 and TGF-β1 pathways [80,81]. The analysis of samples from patients with lung adenocarcinoma shows that CD44 levels are higher in tumors than in normal lung tissue [79], correlate with the expression of EMT markers (inversely with E-cadherin and directly with Snail and Twist) [77,82], and predict the likelihood of metastasis [79].

Collagens play an integrative role in both normal and tumor tissues, connecting different cell types [83,84]. In addition to their structural function, collagens I and XVII have been found to be induced during inflammation and induce EMT in lung cancer cells [85,86,87,88].

Collagen I provides structural stability to the bronchi, alveoli, and interstitium, but its abnormal deposition is associated with respiratory diseases, including asthma, pulmonary fibrosis, and lung cancer [89]. TGF-β upregulates type I collagen in lung tumor cells by inhibiting miR-200 and derepressing ZEB1, and collagen I then induces EMT through the integrin β1/FAK/Src pathway [85,88]. Only mature collagen fibers can induce EMT, as knockdown of the collagen cross-linking enzyme LOXL2, which is also regulated by ZEB1, abolishes FAK and Src activation and the metastasis of mouse lung cancer 344SQ cells in vitro and in vivo [85]. Another key player in this pathway is v-crk sarcoma virus CT10 oncogene homologue (avian)-like (CRKL), which serves as a scaffolding protein in integrin-based focal adhesions. Although CRKL knockdown does not affect EMT markers, it alters cell morphology to round-shaped, decreases migration and invasion, and dysregulates FAK and Src localization in human H157 lung cancer cells [88]. Shintani et al. showed that plating A549 cells on collagen I-coated dishes induces TGF-β3 expression via the PI3K/AKT pathway, thereby forming an autocrine loop to maintain EMT, as evidenced by the reversal of changes in morphology and molecular markers by neutralization with anti-TGF-β3 antibody or PI3K/ERK inhibitors [86]. The analysis of immunohistochemistry (IHC) data from 490 lung cancer samples by Peng et al. showed increased levels of collagen I in poorly differentiated tumors and its correlation with Zeb1 expression, which is consistent with the aforementioned in vitro results. In addition, increased collagen I (*COL1A1*) and *LOXL2* mRNA levels were associated with worse survival in lung adenocarcinoma patients from The Cancer Genome Atlas (TCGA) cohort [85].

Collagen XVII facilitates epithelial cell adhesion to the basement membrane but has also been associated with invasiveness, particularly in lung cancer [90]. The role of collagen XVII in cancer-associated lung inflammation is not fully understood. Autoantibodies to collagen XVII are elevated in the serum of lung cancer patients receiving anti-PD1/PD-L1 therapy and correlate with better responses to therapy and survival, suggesting an important role of collagen XVII expression for T cell effector function in the tumor [91]. On the other hand, collagen XVII induces EMT in lung cancer cells by inhibiting ubiquitin-mediated Snail degradation through the FAK/AKT/GSK3β pathway [87]. Important features of this pathway are the ADAM9/10-mediated shedding of collagen XVII and the subsequent stabilization of laminin-5, which was confirmed by the inhibition of EMT in A549 and CL1-1 cells by short hairpin RNA (shRNA)-mediated knockdown of the corresponding genes [87]. IHC analysis of surgically resected lung tumors from 98 patients showed a negative correlation between survival and positive staining for collagen XVII and laminin-5 [87].

#### 2.2.5. Reactive Oxygen Species in the Regulation of EMT in Lung Cells

The production of reactive oxygen species (ROS) is induced in immune and epithelial cells during the inflammatory response as a defense mechanism against foreign cells such as bacteria or tumor cells. However, oxidative stress has been identified as another factor contributing to inflammation-induced EMT (Figure 2).

MSCs are an established cell type from the iTME that promotes the EMT in lung cancer cells by inducing ROS production. Luo et al. used a co-culture system to show that MSCs increase intracellular ROS in A549 cells, and this induces the EMT through an autophagy-dependent mechanism, as evidenced by autophagosome formation, increases in the autophagic markers Beclin-1 and LC3-II, and the sensitivity of EMT to autophagy modulators (rapamycin, 3-methyladenine, and bafilomycin) [92].

ROS have been recognized as important mediators of TGF-β-induced EMT. TGF-β triggers ROS production in lung cancer cells through the NF-κB-dependent induction of NADPH oxidase 4 (NOX4) expression [93,94] Interestingly, in vitro studies suggest that this pathway is required for the induction of EMT by TGF-β, as pharmacological inhibition of NOX4 or NF-κB suppresses ROS production and reverses changes in motility and EMT markers in A549 cells [93,94]. ShRNA-mediated knockdown of the antioxidant transcription factor NRF2 enhances the response of A549 cells to TGF-β, increasing cell motility, p-SMAD2/3 and NOX4 expression, and intracellular ROS levels, identifying NRF2 as a negative regulator of the TGF-β/NF-κB/NOX4/ROS axis [94]. Another pathway associated with TGF-β-induced ROS production involves increasing the cellular pool of labile iron through ferritin heavy chain (FHS) suppression by transmembrane prostate androgen-induced protein (TMEPAI). ROS then inhibit the EMT suppressor insulin receptor substrate-1 (IRS-1) [95]. Hu et al. showed that TGF-β-induced EMT was inhibited by TMEPAI depletion in A549 cells and could be rescued by either hydrogen peroxide (H_2_O_2_) treatment or small-interfering RNA (siRNA)-mediated knockdown of insulin receptor substrate 1 (IRS-1). Increased expression of TMEPAI in tumors compared to adjacent lung tissue found in 30 patients with squamous cell lung cancer supports the clinical relevance of the above mechanism [95].

#### 2.2.6. COVID-19-Associated Inducers of EMT in Lung Tissue

Given the unprecedented spread and severe consequences of severe acute respiratory syndrome coronavirus 2 (SARS-CoV-2) infection on global public health, the relationship between COVID-19, lung cancer progression, and the EMT requires brief discussion. To date, available clinical studies have clearly demonstrated the high susceptibility of lung cancer patients to SARS-CoV-2 infection [96]. This effect may be related to the overexpression of angiotensin converting enzyme 2 (ACE2) in lung cancer compared to normal tissue, which is a key target for SARS-CoV-2 cell entry [97], and the suppressed immunity of cancer patients [98]. In turn, viral infection may create a microenvironment that enhances the aggressive properties of lung cancer cells, including the induction of the EMT. For example, recent work by Saygideger et al. found that serum samples from COVID-19 patients increased the motility of A549 cells, their loss of intercellular junctions, and the switch in the expression of EMT-related markers to mesenchymal-type markers [99], which is consistent with significant EMT-associated changes in lung lesions in COVID-19 patients who died of the disease reported by Falleni et al. [100]. In addition, a retrospective analysis showed that cancer patients had increased pulmonary metastatic lesions 6 months after SARS-CoV-2 infection [99], while no direct oncogenic effect of this virus has been reported to date [101]. The major inducers of EMT produced by COVID-19 are TGF-β, pro-inflammatory cytokines including IL-6 and IL-1β [102], and the urokinase plasminogen activator (uPA), the latter of which is involved in the production of plasmin, which degrades components of the ECM and activates TGF-β [103]. Furthermore, neutrophil extracellular traps (NETs) produced by neutrophils recruited into SARS-CoV-2-infected lungs have also been implicated in EMT induction in lung cancer cells [9]. Thus, the inflammatory background caused by viral infection may also be involved in the induction of the EMT in lung cancer. More details on the relationship between SARS-CoV-2 infection, tumor progression, and the EMT can be found in recent comprehensive reviews [31,103,104,105].

## 3. Inflammation-Induced EMT as a Source of Cancer Stem Cells

The efficacy of current chemotherapy in the treatment of lung cancer is limited by the acquisition of drug resistance by cancer cells. According to the latest concept, a key role in this process is played by cancer stem cells (CSCs), a population of cancer cells that survive cytotoxic exposure in a quiescent state and then differentiate into heterogeneous types of tumor cells, leading to tumor relapse [106]. The origin of CSCs is still debated, but it is most likely that they arise from proliferating lung epithelial cells capable of transdifferentiation, i.e., the facultative stem cells (alveolar epithelial type II cells (AEC2s) and club cells in the distal lung regions and basal cells in the proximal lung regions) [107].

Cells from the tumor microenvironment, such as CAFs and TAMs, have been shown to provide a supportive niche for CSCs. Chronic inflammation that develops during tumor progression upregulates the stemness of CSCs through activation of the EMT by cytokines produced by inflammatory cells infiltrating the tumor stroma [107]. Macrophages with the M2 phenotype have been shown to play an important role in supporting CSCs through the secretion of anti-inflammatory cytokines TGF-β1 [108,109] and IL-10 [110], but the same activity has also been demonstrated for some pro-inflammatory cytokines such as TNF-α [111], IFN-γ [112], IL-6 [113,114], and IL-17 [115].

M1 and M2 macrophages stimulate the stemness of human H1299 and mouse D121 lung cancer cells by inducing the expression of the deubiquitinase ubiquitin-specific peptidase 17 (USP17), which disrupts the TNFR-associated factor (TRAF)2/TRAF3 complex and thereby inhibits the degradation of its downstream targets NIK, c-Rel, and IRF5, which are involved in the regulation of stemness-related genes [116]. Furthermore, the binding of USP17 to TRAF2/TRAF3 activates the NF-κB signaling pathway [116], which, in addition to regulating the inflammatory response, is involved in the induction of EMT and the acquisition of stem-like properties in NSCLC cells [117]. M2 macrophage-derived IL-10 has been shown to induce stemness in A549 and H460 NSCLC cell lines via the JAK1/STAT1/NF-κB/Notch1 pathway [110], and IL-10 expression by TAMs correlates closely with the NSCLC stage and survival of NSCLC patients [110,118]. Another cytokine secreted by M2 macrophages, TGF-β1, induces EMT in primary lung cancer cells through the downregulation of miR-138 with a subsequent increase in the colony-forming ability and expression of stem cell markers CD44 and CD90 in lung cancer cells [108]. The interaction between TGF-β receptor type II (TβRII) and epithelial membrane protein 3 (EMP3) plays a critical role in the TGF-β1-mediated induction of the EMT and stem-like properties, as evidenced by the reduced migratory ability and resistance to irradiation in EMP3 knockdown A549 cells and the positive correlation between high EMP3 levels and poor survival rates in patients with NSCLC [109].

IL-6 has been reported to increase proliferation and induce EMT in CSC-like A549 and H157 cells expressing the stem cell marker CD133 (CD133+ cells), but not in CD133- cells [113]. Increased CSC proliferation observed upon activation of the IL-6/JAK2/STAT3 signaling axis is associated with the upregulation of DNA methyltransferase I (DNMT1), which decreases the expression of p53 and p21 due to DNA hypermethylation [114]. IL-17 produced by Th17 cells has been shown to increase the migration, invasion, spheroid forming potential, and expression of mesenchymal and stem cell markers in A549 and H460 cells via the STAT3/NF-κB/Notch1 pathway [115]. The effect of IFN-γ on NSCLC cells is dose-dependent: low-dose IFN-γ induces the EMT and enhances the stemness of A549 and H460 cells both In vitro and in vivo by activating the intercellular adhesion molecule 1 (ICAM1)/PI3K/Akt/Notch1 signaling axis; high-dose IFN-γ induces apoptosis of NSCLC cells through the JAK1/STAT1/caspase-3,7 pathway. Analysis of tumor samples from NSCLC patients has shown that low, but not high, levels of IFN-γ and high expression of ICAM1 in the tumor microenvironment are positively correlated with enrichment in CD133+ cells and the highest expression of stemness-related genes [112].

Primary induction of the EMT by inflammatory cytokines stimulates NSCLC cells to secrete soluble factors that act in an autocrine manner to maintain tumor stemness. Wamsley et al. reported that the stimulation of A549 cells with TGF-β and TNF-α induces NF-κB-mediated production of inhibin subunit beta A (INHBA)/Activin, a member of the TGF-β superfamily of growth factors, which further maintains the expression of the EMT TFs Snail, Slug, and ZEB2, as well as the CSC markers N-Myc, SRY-box transcription factor 2 (SOX2), Krüppel-like factor 4 (KLF4), and high mobility group AT-hook 2 (HMGA2) [111]. Furthermore, Activin expression is elevated in tissue samples from patients with primary NSCLC tumors, including adenocarcinoma, squamous cell carcinoma, and large cell carcinoma, further supporting its role as an autocrine factor in maintaining CSC phenotypes [111]. In addition to Activin, the observed effect of TGF-β on the stemness of lung cancer cells appears to be dependent on chemokine (C-X-C motif) ligand 12 (CXCL12), as the upregulation of CXCL12 and its receptor chemokine (C-X-C motif) receptor 7 (CXCR7) is positively correlated with the increase in spheroid-forming potential and CSC marker expression in A549 cells treated with TGF-β. In clinical samples, the co-expression of TGF-β1 and CXCR7 positively correlates with CD44 levels in advanced lung adenocarcinoma, confirming the involvement of CXCL12 in maintaining stem-like properties of NSCLC cells [119]. Wnt3A is another key player in the autocrine/paracrine regulation of cancer cell stemness upon EMT activation. Interestingly, EMT induction alters the set of genes regulated by the Wnt3A/β-catenin pathway by switching from β-catenin/E-cadherin/Sox15 to β-catenin/Twist1/TCF4 complex formation [120]. In mesenchymal-like A549 cells, Twist1 stabilizes β-catenin and enhances the transcriptional activity of the β-catenin/TCF4 complex, leading to an increase in the spheroid-forming capacity and expression of CSC markers. The clinical significance of the Wnt/β-catenin pathway in promoting the CSC phenotype is supported by the analysis of human lung cancer samples, which shows that high-grade primary and metastatic cancers have higher levels of CD133 and nuclear fractions of β-catenin and Twist1, while E-cadherin and Sox15 levels are significantly decreased compared to low-grade primary cancers [120].

## 4. Clinical Trials of Drugs Targeting Key Regulators of Inflammation-Driven EMT

Rigorous study of the relationship between inflammation and tumor progression has led to the development of a wide range of drug candidates targeting key regulators of inflammation-driven EMT, some of which are now in various stages of clinical trials. Surprisingly, despite the encouraging tumor suppressive effects of TGF-β-targeted compounds in vitro and in vivo, their antitumor efficacy in oncology patients is mainly controversial with a small survival benefit [121]. For example, trabedersen (an antisense oligonucleotide targeting TGF-β2), Lucanix (a non-viral gene-based allogenic tumor cell vaccine targeting TGF-β2), and the combination of galunisertib (TβRI kinase inhibitor) and lomustine showed no significant antitumor activity in clinical trials in patients with brain tumors (phase IIb, NCT00431561) [122], NSCLC (phase III, NCT00676507) [123], and recurrent glioblastoma (phase II, NCT01582269) [124], respectively. Nevertheless, a number of studies still showed favorable survival outcomes in cancer patients after anti-TGF-β therapy, suggesting a possible relationship between the efficacy of TGF-β-targeting drugs and the tumor context and the level of their bioavailability. For example, galunisertib was shown to significantly enhance the antitumor effect of sorafenib in patients with hepatocellular carcinoma (phase II, NCT01246986) [125], and its combination with temozolomide-based chemoradiation improved overall survival in patients with glioblastoma (phase I/II, NCT01220271) [126]. In addition, the phase III clinical trial (NCT00761280) showed that trabedersen increased the two-year survival in patients with secondary glioblastoma and anaplastic astrocytoma to 35.7% compared with 23.1% for conventional chemotherapy [126]. The anti-TGF-β1-3 monoclonal antibody called fresolimumab (also known as GC1008) at 10 mg/kg in combination with radiation was found to cause a significantly longer median overall survival with better immunologic parameters in patients with metastatic breast cancer compared to fresolimumab administration at 1 mg/kg (Phase II, NCT01401062) [127].

In addition to anti-TGF-β therapy, the inhibition of a number of pro-inflammatory mediators has also conferred a favorable survival benefit in cancer patients. For example, the use of celecoxib (a selective COX-2 inhibitor) as an adjuvant to chemotherapy significantly improved the overall survival and disease-free survival in COX-2-positive gastric cancer patients [128], and its oral administration significantly improved responses in patients with recurrent ovarian cancer characterized by platinum-based drug resistance (phase II, NCT01124435) [129]. Another COX-2 inhibitor, rofecoxib, in combination with cyclophosphamide and vinblastine, showed a 30% clinical benefit in patients with advanced solid tumors [130], and low-dose aspirin prevented colorectal cancer in patients with familial adenomatous polyposis by suppressing the recurrence of colorectal polyps (UMIN000018736) [131], a process associated with EMT [132]. These results are consistent with the recent retrospective analysis by Cai et al. demonstrating that the use of non-steroidal anti-inflammatory drugs during radical resection in NSCLC patients correlates with outcomes [133]. In addition, the blockade of IL-1 signaling has demonstrated significant antitumor efficacy in clinical trials. Treatment of patients with advanced colorectal cancer with bermekimab (a monoclonal antibody against IL-1α) was found to significantly improve the clinical response rate (phase III, NCT01767857), and nadunolimab (a monoclonal antibody against IL-1 receptor accessory protein (IL1RAP)) in combination with nab-paclitaxel and gemcitabine demonstrated a clinical benefit rate of 74% (phase I/IIa, NCT03267316) [134]. Consistent with this, an additional analysis of the Canakinumab Anti-Inflammatory Thrombosis Outcomes Study (CANTOS) trial in 10,061 patients showed that the administration of canakinumab (a monoclonal antibody against IL-1β) dose-dependently reduced the incidence of lung cancer and lung cancer mortality (phase III, NCT01327846) [135].

Taken together, the described clinical data, although not directly indicative of blockade of inflammation-driven EMT in cancer patients, clearly confirm the stimulatory effect of iTME on tumor progression and metastasis. The demonstrated clinical benefit for cancer patients in response to anti-TGF-β and anti-inflammatory drugs suggests the prospect of further investigation of the “inflammation—tumor growth” relationship, including the induction of EMT in tumor cells by an inflammatory background.

## 5. Inhibitory Effect of Triterpenoids on Inflammation-Induced EMT in Lung Cancer Cells

Natural metabolites represent an important source of drug candidates with pronounced anti-inflammatory and anti-tumor properties [10]. These compounds can be considered effective inhibitors of the EMT in lung cancer cells, as has already been shown for lignans (arctigenin [136]), alkaloids (berberine [137]), flavonoids (luteolin [138]), phenolic compounds (eugenol [139]), resveratrol [140], curcumin [141]), and other molecules (more details on this topic can be found in the following recently published comprehensive reviews [142,143,144]). Given the lack of reviews analyzing in detail the bioactivity of pentacyclic triterpenoids (PTs) against the links between inflammation and EMT, this chapter will focus on this class of natural compounds.

PTs are terpenes whose carbon backbone consists of six isoprene units. They can be isolated from various sources such as fungi and marine organisms, but most bioactive PTs have been obtained from plants. In plants, the mevalonate and methylerythritol phosphate pathways synthesize isopentenyl pyrophosphate (IPP), some of which is isomerized to dimethylallyl pyrophosphate (DMAPP). Two IPP units are then added to DMAPP to form farnesyl pyrophosphate (FPP), which is dimerized to form squalene. Squalene epoxidase oxidizes squalene to 2,3-oxidosqualene, the common precursor of all PTs, after which the metabolic pathways diverge to produce different types of triterpenoid scaffolds (Figure 5) [145].

From a biological point of view, the most important types of PTs are lupane, oleanane, ursane, and friedelane. Lupane-type PTs have a five-carbon E-ring. In contrast, oleanane and ursane E-rings are six-membered and can be distinguished by the position of the methyl group attached to the C-20 and C-19 atoms, respectively. The friedelane-type backbone is derived from the oleanane type by methyl translocation [145] (Figure 5). Representative sources of PTs are *Betula pubescens* (white birch) for lupane [146], *Malus domestica* (apple), *Coffea arabica* (arabic coffee), and *Origanum vulgare* (oregano) for ursane [147], *Glycyrrhiza glabra* (licorice), *Olea europaea* (olive), and *Evodia rutaecarpa* (evodia) for oleanane [148,149,150], and *Tripterygium wilfordii* and *Celastrus orbiculatus* (oriental bittersweet) for friedelane [151,152]. However, the diversity of PTs is not limited to these species; they can be found in many food and medicinal plants.

PTs have a variety of biological activities, including lipid-lowering [153], anti-diabetic [154], antidepressant [155], antinociceptive [156], anti-inflammatory [157], and antitumor [158] effects. PTs inhibit the immune response by reducing the production of ROS, nitric oxide (NO), and proinflammatory cytokines by macrophages [157]. In tumors, PTs induce cell death via autophagy and apoptosis and inhibit metastasis and neoangiogenesis [158,159]. The ability to inhibit EMT has been reported for all types of PTs [160,161,162,163], but these studies have not paid special attention to inflammation-driven EMT. Therefore, the next part of our study focuses on the analysis of available published data dedicated to the modulating effect of PTs on EMT induced by inflammatory background in lung cancer. The results of this analysis are summarized in Figure 6 and Table 1.

### 5.1. Ursane-Type Triterpenoids

Ursolic acid (UA), a compound synthesized in a variety of plants, has been shown to inhibit basement membrane adhesion, migration, invasion, and EMT of A549 cells via NF-κB-dependent downregulation of astrocyte-elevated gene-1 (AEG-1) [166]. A study by Ruan et al. demonstrated that UA inhibited TGF-β-induced EMT in H1875 cells through the downregulation of the Smad-independent integrin αVβ5 pathway [167]. However, the canonical Smad-dependent pathway is also responsive to UA treatment, as evidenced by the correlation between reduced levels of TGF-β and phosphorylated Smad3 and the attenuation of EMT and airway-vessel remodeling in the lungs of UA-treated Wistar rats with cigarette smoke-induced emphysema. In addition, UA induced the expression of insulin-like growth factor 1 (IGF-1) [168], which has been suggested to play a key role in the regeneration of epithelial and muscle cells in patients with COPD [181]. Lin et al. also demonstrated that UA effectively blocks the unfolded protein response (UPR) signaling cascades that cause the alleviation of emphysema and airway remodeling in the lungs of Sprague–Dawley rats induced by the administration of cigarette smoke extract [169].

Another ursane-type triterpenoid, asiatic acid (AA), extracted from *Centella asiatica* (L.), also inhibited TGF-β-induced EMT of A549 cells [164]. This effect was further confirmed in a bleomycin-induced pulmonary fibrosis model in C57BL/6 mice; the administration of AA for 21 days resulted in the alleviation of pulmonary fibrosis by blocking EMT through the downregulation of TGF-β1/Smad2/3 and ERK1/2 pathways. Interestingly, along with the reversal of EMT, AA suppressed lung inflammation by reducing inflammatory cell infiltration in bronchoalveolar lavage fluid and the expression of pro-inflammatory cytokines (IL-1β, IL-18, IL-6, and TNF-α) in lung tissue as well as inhibiting IL-1β and IL-18 secretion via the inactivation of NLRP3 inflammasome [165]. Considering the above-mentioned EMT-inducing activity of these cytokines, AA-mediated inhibition of EMT in the lungs of mice with pulmonary fibrosis may be associated with the inhibition of TGF-β signaling pathways, but also with the inhibition of some other signaling axes susceptible to EMT-inducing cytokines.

### 5.2. Oleanane- and Friedelane-Type Triterpenoids

Oleanolic acid, a plant oleanane-type triterpenoid, can inhibit EMT in the human NSCLC cell lines A549 and PC-9 through the ERK pathway [175]. A number of other oleanane-type triterpenoids have been reported to inhibit the EMT in lung epithelial cells, primarily through the inhibition of the TGF-β1/Smad2/3 pathway. Celastrol isolated from *Tripterygium wilfordii Hook F.* inhibited TGF-β1-induced EMT in A549 cells [152]. These results are supported by in vivo studies showing that celastrol treatment suppressed the activation of the TGF-β1/Smad2/3 pathway in the lungs of Wistar rats with bleomycin-induced pulmonary fibrosis. As a result, the EMT was inhibited as evidenced by an increase in the expression of epithelial markers (E-cadherin and claudin) and suppression of the expression of mesenchymal markers (N-cadherin, β-catenin, Snail, and Slug) in rat lung tissues, respectively. These effects of celastrol were partly due to the downregulation of Hsp90 expression [174], which is a negative regulator of E-cadherin expression [182]. Like celastrol, β-peltoboykinolic acid isolated from *Astilbe rubra* also inhibits the EMT in A549 cells through the TGF-β1/Smad2/3 pathway [178].

Gui et al. reported that HMGB1 downregulation underlies glycyrrhizin-mediated reversal of the EMT in TGF-β1-treated A549 and BEAS-2B cells, as HMGB1 has been shown to promote TGF-β1-induced Smad2/3 phosphorylation [149]. Additionally, Ren et al. found that the HMGB1-blocking effect of glycyrrhizin determines its suppressive effect on the CAF-driven highly motile and invasive phenotype of A549 and H661 cells [40].

Glycyrrhizin is the parent compound in the synthesis of soloxolone methyl (SM), a triterpenoid with a cyano-enone pharmacophore group in its structure [183], which reduced migratory and invasive abilities and inhibited the EMT in TGF-β1-stimulated A549 cells. The analysis of the gene association network consisting of previously established EMT-associated genes and in silico-predicted molecular targets of SM with further verification by molecular docking approach revealed that SM-mediated EMT inhibition could be due to its interactions with MMP-2, MMP-9, and mitogen-activated protein kinase 8 (MAPK8), but further experimental verification is needed to prove the interaction between SM and its targets [176]. 2-cyano-3, 12-dioxooleana-1, 9-dien-28-oic acid (CDDO), a structural analog of SM synthesized from oleanolic acid, has been shown to attenuate radiation-induced lung inflammation and fibrosis in C57BL/6 mice. CDDO inhibited inflammatory cell infiltration, thereby reducing the levels of the EMT-inducing cytokines TGF-β and IL-6 produced by these cells in the lungs of X-ray-treated mice. In addition, CDDO inhibited the expression of the mesenchymal markers fibronectin, α-SMA, and collagen I, as well as the deposition of collagen fibers in the lungs of irradiated mice, suggesting that CDDO inhibits the EMT in lung epithelial cells in mice with radiation-induced lung fibrosis [173]. Pristimerin isolated from *Celastraceae* plants and evoditrilone A isolated from *Evodia rutaecarpa* were reported to inhibit migration ability and regulate EMT-associated markers in H1299 and A549 cells, respectively [148,151]. The exact mechanisms of the EMT-inhibitory activity of cyanoenone-bearing PTs, as well as pristimerin and evoditrilone A, remain to be elucidated.

### 5.3. Lupane-Type Triterpenoids

Among the lupane triterpenoids, betulinic acid (BA) inhibited TGF-β1-induced EMT in H1299 and A549 cells in vitro and metastasis in a variety of mouse models in vivo. He et al. demonstrated that BA inhibits NSCLC metastasis by direct interaction with Skp2, thereby blocking the assembly of the Skp2-SCF E3 ligase complex, which causes impairment of the ubiquitin-dependent degradation of its downstream target E-cadherin [179]. The introduction of a 4-methoxyphenylacetic group at the C-3 position of BA (a derivative named SYK023) significantly improved its inhibitory effect on the migration and invasive ability of H1299 in vitro and lung metastasis in nude mice bearing CL1-5 human lung cancer xenografts in vivo. An additional mechanism contributing to the antimetastatic activity of BA and SYK023 was reported by Hsu et al. It was shown that these compounds effectively blocked F-actin polymerization by inhibiting the Src/FAK and Akt/mTOR pathways and suppressing the expression of many actin polymerization genes, among which synaptopodin (Synpo) plays a key role, as its knockdown prevented the inhibitory effect of BA and SYK023 on cell motility and F-actin polymerization [180]. These effects may be related to the established role of Synpo in blocking Smurf1-mediated ubiquitination and degradation of RhoA, a key regulator of the actin cytoskeleton [184]. In addition, both BA and SYK023 were shown to inhibit the expression of N-cadherin, vimentin, and β-catenin in H1299 cells, which is in good agreement with their anti-EMT effect demonstrated by He et al. [179,180].

## 6. Inhibitory Effect of Triterpenoids on Stem-like Properties of Lung Cancer Cells

As previously described, chronic inflammation that develops during cancer progression induces the transformation of normal tissue stem cells into cancer stem cells (CSCs), which have the ability to self-renew and differentiate into cancer cells, thus contributing to tumor recurrence and drug resistance. Because EMT plays a critical role in maintaining the stem-like properties of CSCs, targeting EMT pathways could potentially alter the response of CSCs to anticancer drugs and reduce the probability of tumor relapse after chemotherapy.

UA has been shown to inhibit tumorsphere formation and expression of the CSCs markers NANOG, OCT4, and SOX2 in A549 and H460 cells by targeting the activation of EGFR and its downstream JAK2/STAT3 pathway [170] (Figure 7). A similar effect on the sphere-forming potential of A549 and H1299 cells was demonstrated for BA, but the mechanism of its inhibitory effect on the stem-like properties of NSCLC cells differs from that of UA and seems to be related to the above-mentioned direct interaction of BA with Skp2 and blocking the assembly of the Skp2-SCF complex, which inhibits the degradation of the cyclin-dependent kinase inhibitor p27 and thus compromises CSCs proliferation [179,185].

Aldehyde dehydrogenase 1A1 (ALDH1A1) has been shown to promote the stemness of CSCs by catalyzing the conversion of retinol to retinoic acid, which induces the expression of stem cell-related genes either through the classical pathway, in which retinoic acid binds to the nuclear retinoic acid receptor α (RARα)/retinoid X receptor (RXR) heterodimer, or through the alternative pathways, including activation of the PI3K/Akt pathway via binding to cytosolic RARα [186]. β-Escin, a triterpenoid saponin isolated from *Aesculus hippocastanum*, was shown to decrease ALDH activity and ALDH1A1 expression in H460 cells in vitro and eradicate the population of ALDH1A1+ cells in lung adenocarcinomas induced by the tobacco carcinogen 4-(methyl nitrosamino)-1-(3-pyridyl)-1-butanone (NNK) in A/J mice in vivo. As a result, there was a marked decrease in Akt activation in the lung tumors of β-Escin-treated mice, confirming the relationship between the inhibitory effect of BA on CSCs proliferation and the suppression of retinoic acid-activated signaling pathways [177].

## 7. Future Prospective and Limitations

An increasing number of studies indicate that the inflammatory microenvironment established during tumor progression promotes lung cancer metastasis through the induction of EMT in tumor cells. A number of inflammation-related cytokines produced by various cells in iTME have pleiotropic effects and, in addition to their immunomodulatory potency, are able to stimulate EMT in lung cancer cells. The invasive capacity of cancer cells increases as they undergo EMT due to cytoskeletal rearrangements and the production of enzymes that degrade ECM components. In addition, recent studies have shown a link between EMT induction and the formation of lung CSCs, a population of stem-like cancer cells involved in the acquisition of chemoresistance and tumor recurrence. Therefore, targeting iTME-mediated induction of EMT with chemotherapeutic agents may be a promising strategy to suppress metastasis and drug resistance in lung cancer, as evidenced by the clinical benefit for cancer patients in response to drugs targeting iTME components in clinical trials.

PTs of the oleanane-, ursane-, and lupane-type exhibit suppressive potency against inflammation-induced EMT in lung cancer cells. It should be emphasized that in the majority of published studies on this topic, researchers have described the effect of PTs mainly on the TGF-β signaling pathway, while pathways sensitive to other EMT-associated regulators remained outside their attention and require further detailed investigation. The demonstrated ability of some PTs to markedly inhibit the expression of inflammation-related cytokines in the lungs of mice with pulmonary fibrosis, accompanied by modulation of the expression of EMT markers, suggests the prospect of further, more detailed studies of the effect of PTs on cytokine-driven EMT in lung cancer cells.

Since the inflammatory process is characterized by the production of a large pool of cytokines acting synergistically, co-culture models of lung cancer cells with activated immune cells are of great interest in this direction. It should be noted that the majority of published in vitro experiments on the anti-EMT potency of PTs have been performed using very simple cell models activated by EMT inducers only in a monotherapy regimen. Co-culture studies would allow a more thorough comparison of the data obtained from in vitro and in vivo experiments and allow more objective conclusions about the mechanism of anti-EMT action of the studied compounds.

Despite the currently active research on the relationship between EMT and CSCs, there are only fragmentary data on the effect of PTs on the population of lung CSCs enriched in inflammatory conditions due to EMT induction, and therefore it is not possible to fully describe the molecular mechanism underlying stemness inhibition by PTs. However, the paucity of studies on this topic may not be due to the lack of CSC-inhibitory activity of PTs, but to the fact that researchers have overwhelmingly focused on the effect of compounds on the motility and invasiveness of EMT-derived lung cancer cells in the context of their antimetastatic activity, overlooking their effect on stem-like properties. Thus, the study of the effect of PTs with previously confirmed anti-EMT activity, as well as newly developed triterpenoids, on the stemness of lung cancer cells is of great interest for further characterization of the molecular mechanism of their antitumor activity.

## 8. Conclusions

Taken together, the data presented in this review suggest a clear link between inflammation and the induction of the EMT in tumor cells and, as a consequence, the enhancement of their malignancy, which opens new opportunities for the development of antitumor agents. PTs exhibiting pronounced anti-inflammatory and antitumor potential can be considered a promising source of effective blockers of inflammation-driven EMT in lung cancer cells, but research in this direction is currently at an early stage and requires further development. The information on key inflammation-related signaling pathways associated with the EMT in lung cells described in this work can be used as a kind of roadmap for researchers developing new phytochemical compounds targeting processes closely associated with the EMT, such as cancer and fibrosis.

## Figures and Tables

**Figure 1 ijms-24-17325-f001:**
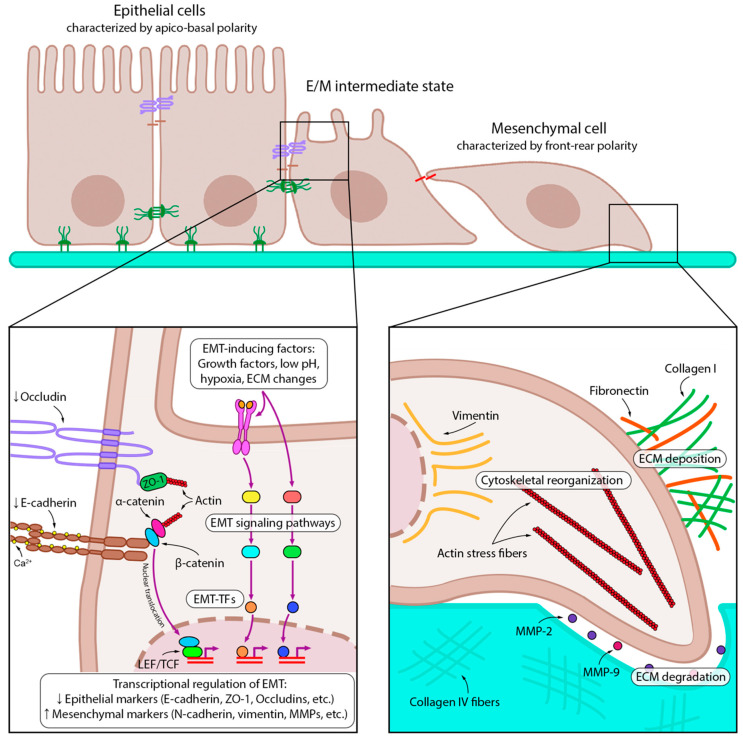
Key molecular events of EMT. Epithelial tumor cells receive EMT-inducing signals from the tumor microenvironment, such as growth factors, low pH, hypoxia, and ECM modifications. EMT-inducing factors trigger various intracellular signaling pathways and activate EMT-TFs, which then downregulate and upregulate epithelial and mesenchymal markers, respectively. EMT is further sustained by autocrine loops: for example, inhibition of E-cadherin leads to its dissociation from β-catenin, which translocates to the nucleus and transactivates EMT-associated genes. Later stages of EMT involve transformation of both intra- and extracellular compartments. The formation of actin stress fibers and vimentin intermediate filaments provides the mechanical force for migration. The cells that have undergone EMT have multidirectional effects on the ECM: they disrupt the basement membrane by degrading collagen IV with matrix metalloproteinases (MMPs) 2 and 9, but at the same time produce other ECM components such as collagen I and fibronectin, which further maintain EMT. Downward (↓) and upward (↑) arrows indicate downregulation and upregulation of expression, respectively.

**Figure 2 ijms-24-17325-f002:**
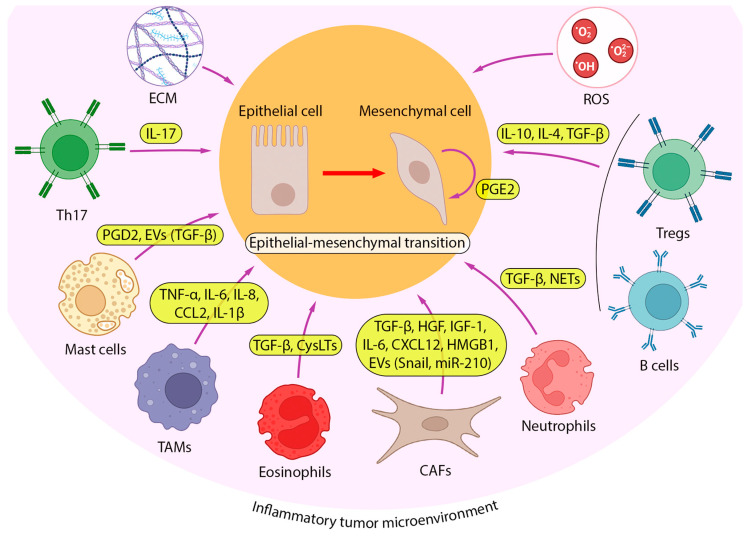
iTME as a driving source of EMT in lung cancer. A number of cell types from iTME, including cancer-associated fibroblasts (CAFs), tumor-associated macrophages (TAMs), neutrophils, eosinophils, mast cells, T cells (T helper 17 (Th17) and regulatory T (Tregs) cells), and B cells, induce EMT in lung cancer cells by producing various cytokines. Among them, transforming growth factor (TGF-β) is the most studied, but EMT-inducing activity has also been reported for anti-inflammatory cytokines (interleukin 4 (IL-4), IL-10), pro-inflammatory cytokines (IL-1β, IL-6, IL-17, tumor necrosis factor α (TNF-α)), chemokines (C-C Motif Chemokine Ligand 2 (CCL2), C-X-C Motif Chemokine Ligand 12 (CXCL12), IL-8), prostaglandins D2 and E2 (PGD2, PGE2), and neutrophil extracellular traps (NETs). Along with secretion, some EMT-inducing factors are transported to tumor cells via extracellular vesicles (TGF-β, Snail, microRNA-210 (miR-210)). In addition, lung tumor cells produce PGE2 to stimulate their EMT in an autocrine manner.

**Figure 3 ijms-24-17325-f003:**
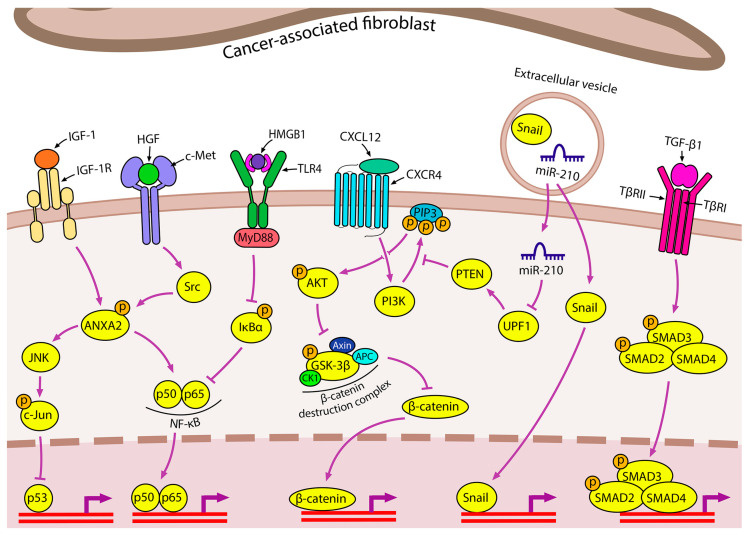
Contribution of CAFs to inflammation-induced EMT in lung cancer CAFs secrete a variety of EMT-inducing factors, including insulin-like growth factor 1 (IGF-1), hepatocyte growth factor (HGF), high mobility group box 1 (HMGB1), CXCL12, TGF-β, and extracellular vesicles (EVs). The IGF-1 and HGF signaling pathways are dependent on annexin A2 (ANXA2) phosphorylation. Autophagy-induced secretion of HMGB1 by CAFs induces EMT via the nuclear factor kappa B (NF-κB) pathway. CXCL12 triggers CXCR4/β-catenin to upregulate EMT-associated genes, among which peroxisome proliferator activated receptor delta (PPARδ) plays a specific role. CAF-derived EVs carry the EMT-TF Snail and miR-210, which induces the PTEN/PI3K/AKT pathway through UPF1 inhibition.

**Figure 4 ijms-24-17325-f004:**
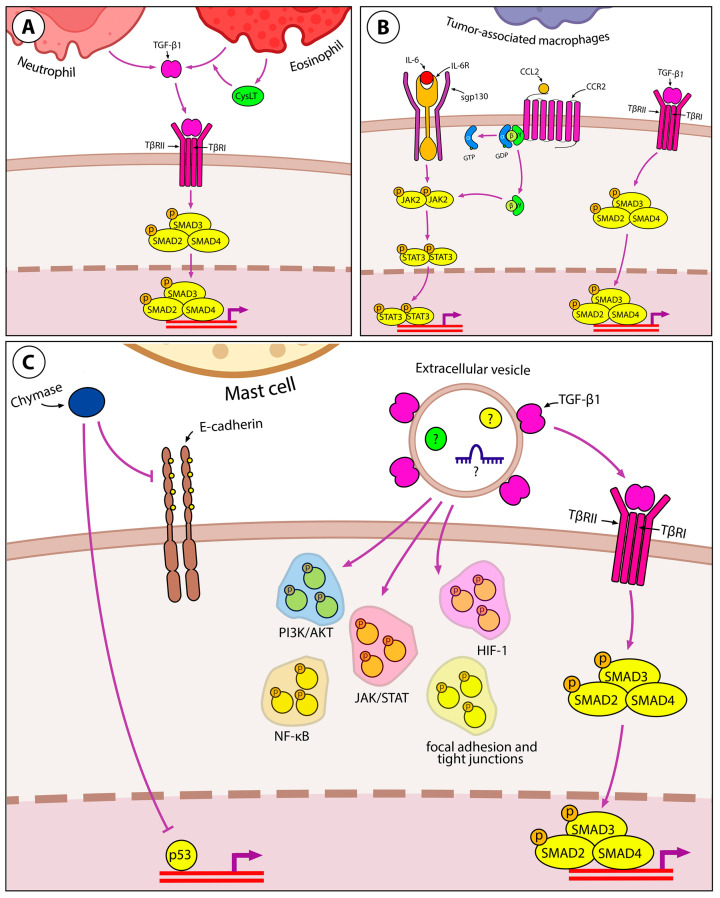
Immune component of inflammation-induced EMT in lung cancer. (**A**) Granulocytes, namely neutrophils and eosinophils, induce EMT through TGF-β/Smad signaling. CysLTs increase TGF-β production in eosinophils in an autocrine manner. (**B**) TAMs induce EMT in lung cancer by releasing IL-6, CCL2, and TGF-β. IL-6 and CCL2 share the same downstream JAK2/STAT3 signaling and mutually upregulate each other to induce EMT. (**C**) Mast cells release chymase, which reduces cell–cell adhesion by cleaving E-cadherin. Inhibition of the p52 tumor suppressor can be mentioned as another process induced by chymase in lung cancer cells. Mast cells secrete TGF-β-coated EVs that induce the classical Smad-dependent pathway. However, EVs contain other EMT-inducing molecules as they activate the phosphorylation of many proteins in lung cancer cells involved in PI3K/AKT, JAK/STAT, NF-κB, and HIF-1 signaling pathways, as well as the formation of focal adhesions and tight junctions.

**Figure 5 ijms-24-17325-f005:**
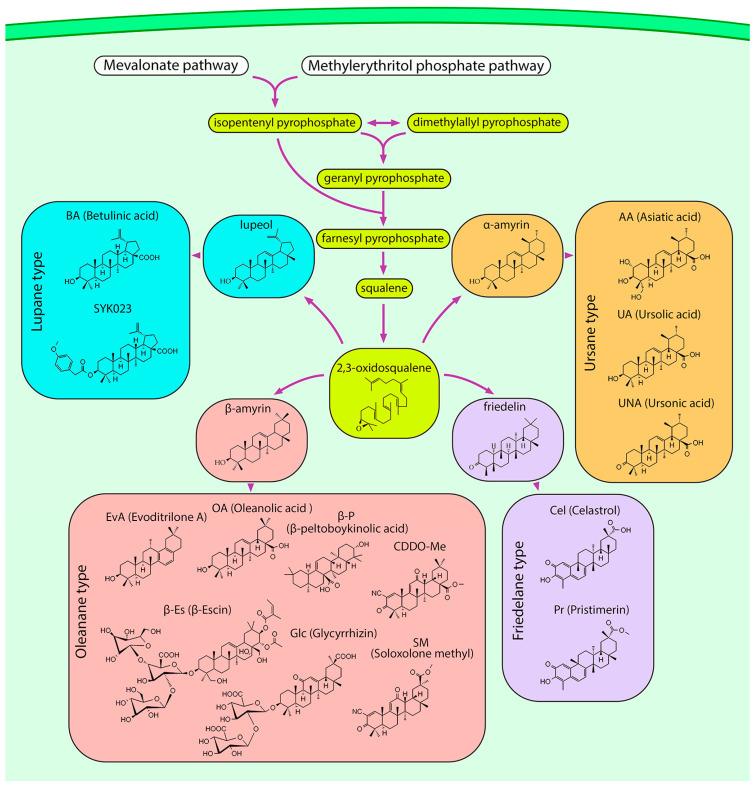
Biosynthetic pathway of PTs. In plant cells, the mevalonate and methylerythritol phosphate pathways provide isoprene units in the form of isopentenyl pyrophosphate (IPP) and dimethylallyl pyrophosphate (DMAPP), which are used as building blocks for the production of squalene. Oxidation of squalene yields 2,3-oxidosqualene, which serves as a precursor for the synthesis of all types of PT scaffolds, including lupane, ursane, oleanane, and friedelane. For simplicity, enzymes and most synthesis steps are omitted.

**Figure 6 ijms-24-17325-f006:**
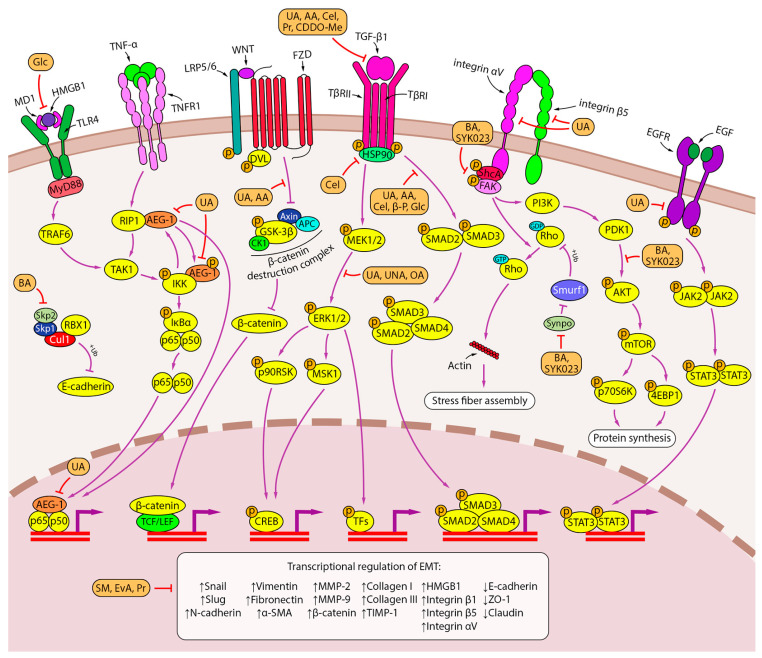
Inhibition of inflammation-induced EMT by PTs. Lupane-, oleanane-, and ursane-type PTs exhibit a variety of EMT inhibition mechanisms. PTs inhibit TGF-β signaling pathways (UA, AA, celastrol, pristimerin, C DDO-Me) and the downstream SMAD (UA, AA, celastrol, β-peltoboykinolic acid, glycyrrhizin) and ERK (UA, UNA, OA) signaling axes. PT-induced disruption of integrin signaling involves integrin αVβ5 expression inhibition (UA) and activation of FAK and AKT (BA, SYK023). The blockade of F-actin polymerization by the lupane-type PTs BA and SYK023 is mediated by the downregulation of Synpo, which presumably activates Smurf1-dependent ubiquitination of RhoA. In contrast, BA disrupts the Skp2-SCF E3 ligase complex, thereby protecting E-cadherin from degradation. The ursane-type PTs UA and AA inhibit β-catenin through a GSK-β-dependent mechanism. The effect of UA on EMT is also associated with repression of the AEG-1 oncogene, which regulates several steps in the NF-κB signaling pathway, and inhibition of the EGFR signaling pathway. Glycyrrhizin inhibits EMT induced by HMGB1, a nuclear protein released from tumor cells (TLR4 and RAGE are shown as possible HMGB1 receptors due to uncertainty in the downstream pathway). Some triterpenoids have an unknown mechanism of action but regulate EMT-associated genes (SM, evoditrilone A, pristimerin). Downward (↓) and upward (↑) arrows indicate downregulation and upregulation of expression, respectively.

**Figure 7 ijms-24-17325-f007:**
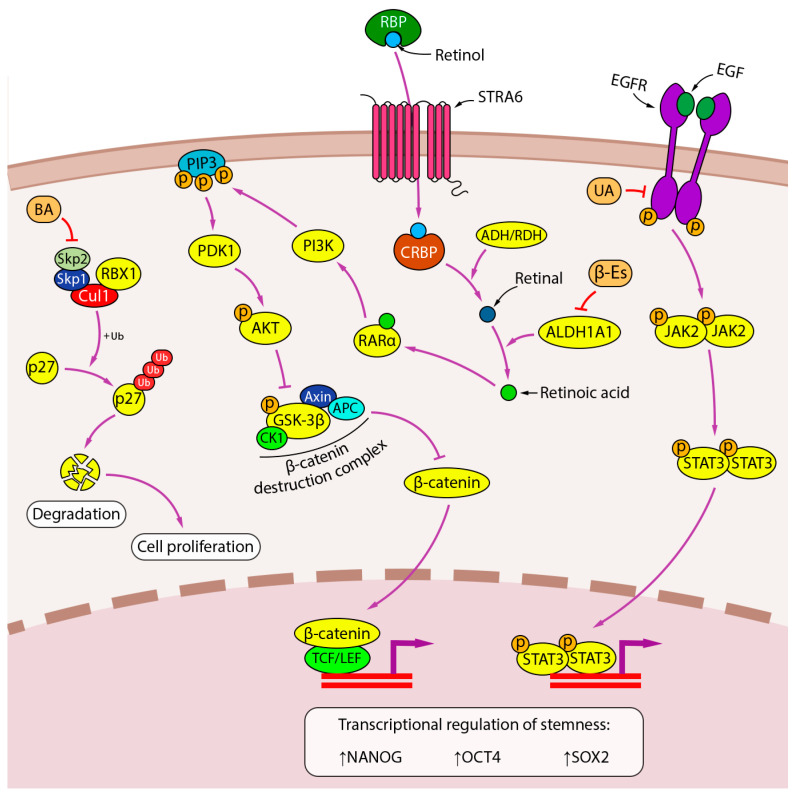
Inhibition of stemness by PTs. Lupane-type PT BA inhibits stem cell proliferation by interfering with Skp2-SCF-dependent degradation of the cyclin-dependent kinase inhibitor p27. Oleanane-type PT β-escin suppresses the activity of ALDH1A1 and the downstream AKT pathway, which is thought to regulate the CSC transcriptome through the β-catenin pathway. The anti-stem cell activity of ursane-type PT UA is mediated through the EGFR pathway. Downward (↓) and upward (↑) arrows indicate downregulation and upregulation of expression, respectively.

**Table 1 ijms-24-17325-t001:** Studies on the inhibition of EMT and stemness by PTs.

PT		Experimental Model	Conditions		Biological Effects ***	Effects on Cell Signalings ****	Refs.
Type	Name		Conc. *	Time **			
Ursane	Asiatic acid (AA)	A549 treated with TGF-β1 (10 ng/mL)	10–40	24 h	↓ morphological changes (20 μM AA), ↓ migration ^DD^, ↓ invasion ^DD^	↓ β-catenin ^DD^, ↓ p-GSK-3β ^DD^, ↑ E-cadherin ^DD^, ↓ N-cadherin ^DD^, ↓ vimentin ^DD^, ↓ Snail ^DD^	[164]
		Pulmonary fibrosis in C57BL/6 mice was induced by single intratracheal administration of bleomycin (3 mg/kg)	5–20 mg/kg/day (intragastrically)	21 d	↓ collagen deposition ^DD^, ↓ histopathological changes in the lungs ^DD^, ↑ pulmonary function ^DD^, ↓ macrophage, neutrophil and lymphocyte infiltration in BALF ^DD^	↓ collagen I ^DD^, ↓ collagen III ^DD^, ↓ α-SMA ^DD^, ↓ TIMP-1 ^DD^, ↓ vimentin ^DD^, ↑ E-cadherin ^DD^, ↓ TGF-β1 ^DD^, ↓ p-Smad2/3 ^DD^, ↓ p-ERK1/2 ^DD^, ↓ IL-1β ^DD^, ↓ IL-18 ^DD^, ↓ IL-6 ^DD^, ↓ TNF-α ^DD^, ↓ NLRP3 ^DD^, ↓ ASC ^DD^, ↓ pro-Caspase-1 ^DD^, ↓ active Caspase-1 ^DD^	[165]
Ursane	Ursolic acid (UA)	A549, H1975	5–30	24 h	↓ adhesion to Matrigel ^DD^ (A549), ↓ migration ^DD^ (A549, H1975), ↓ invasion ^DD^ (A549, H1975)	↑ E-cadherin ^DD^ (A549), ↓ N-cadherin ^DD^ (A549), ↓ vimentin (A549, 20 μM UA); 84 genes regulated by UA (A549, 30 μM UA) were associated with the signaling pathways of TGF-β, ECM-receptors, adherens junctions, Wnt, VEGF, tight junctions, cell adhesion molecules; ↓ AEG-1 ^DD,K^ (A549)	[166]
		A549 treated with TNF-α (5 ng/mL)	5–20	12/24 h with UA, then 12/24 h with TNF-α		↓ NF-κB p65 subunit^K^ (20 μM UA, 12 h), ↓ AEG-1 ^DD^ (24 h)	
		H1975 treated with TGF-β1 (5 ng/mL)	0.02	24 h	↓ morphological changes, ↓ migration, ↓ invasion	↑ E-cadherin, ↓ N-cadherin, ↓ MMP-2 catalytic activity, ↓ MMP-9 catalytic activity, ↓ MMP-2, ↓ MMP-9, ↓ integrin αVβ5^K^	[167]
		HBE treated with 1% cigarette smoke extract (CSE)	10	2 h prior to CSE		↓ TGF-β1, ↓ p-Smad2/3, ↓ S100A4, ↑ IGF-1	[168]
		Emphysema in Wistar rats was induced by exposure to cigarette smoke for 30 min, two times a day, 6 days a week, for 3 months	10–40 mg/kg/day (intragastrically)	3 mos	↓ airway-vessel remodeling, ↓ collagen deposition, ↓ mucus secretion	↓ α-SMA ^DD^, ↓ S100A4, ↓ TGF-β1, ↓ p-Smad2/3, ↑ IGF-1 ^DD^	
		Emphysema in SD rats induced by exposure to cigarette smoke for 30 min, two times a day, 6 days a week, for 3 months	10–40 mg/kg/day (intragastrically)	3 mos		↓ p-IRE1, ↓ XBP1	[169]
		Emphysema in SD rats induced by intraperitoneal injection of CSE on days 1, 8, 15, 21	20 mg/kg/day (intragastrically)	2–4 wk	↓ airway remodeling	↓ p-Smad2/3, ↓ p-PERK, ↓ PERK, ↓ p-eIF-2α, ↓ e-IF-2α, ↓ ATF4, ↓ CHOP, ↓ p-IRE1, ↓ ATF6, ↓ active Caspase 12	
		A549, H460	10, 20	1–14 d	↓ migration ^DD^, ↓ invasion ^DD^, ↓ tumorsphere formation (20 μM UA; 7 d, 14 d)	↓ VEGF ^DD^ (24 h), ↓ NANOG (tumorpheres; 20 μM UA; 24 h), ↓ OCT4 (tumorpheres; 20 μM UA; 24 h), ↓ SOX2 (tumorpheres; 20 μM UA; 24 h), ↓ pEGFR (24 h), ↓ pJAK2 ^DD^ (24 h), ↓ pSTAT3 ^DD^ (24 h), ↓ PD-L1 ^DD^ (24 h), ↓ MMP2 ^DD^ (24 h), ↓ MMP3 ^DD^ (24 h), ↓ MMP9 ^DD^ (24 h), ↓ VEGF ^DD^ (24 h), ↓ the binding of STAT3 to MMP2 and PD-L1 promoters (20 μM UA, 24 h)	[170]
		A549 and H460 treated with EGF (25 ng/mL)	20	1 h with EGF, then 24 h with UA		↓ pEGFR	
		H1975 harbouring L858R/T790M mutation	25	12–72 h	↓ migration	↓ CT45A2^K^ (12 h), ↓ the binding of TCF4 to CT45A2 promotor^K^ (12 h), ↓ TCF4 (48 h), ↓ p-β-catenin (48 h), ↑ p-GSK-3β (48 h), ↓ nuclear translocation of β-catenin	[171]
		subcutaneous injection of H1975 in athymic nude mice	25 mg/kg/day	18 d		↓ CT45A2, ↓ p-β-catenin, ↑ p-GSK-3β, ↓ TCF4	
Ursane	Ursonic acid	A549, H1299	2.5, 5	24 h, 48 h	↓ invasion (24 h)	↓ MMP-2 catalytic activity ^DD^ (A549, H1299; 48 h), ↓ MMP-9 catalytic activity (H1299, 48 h), ↓ MMP-2 ^DD^ (A549, H1299; 48 h), ↓ MMP-9 ^DD^ (H1299, 48 h), ↓ RECK ^DD^ (A549, 48 h), ↑ RECK (H1299, 48 h), ↓ p-ERK ^DD^ (A549, H1299; 24 h), ↓ p-CREB ^DD^ (A549, H1299; 24 h)	[172]
Oleanane	C DDO-Me	Radiation-induced lung inflammation in C57BL/6 mice was induced by thoracic irradiation with a single dose of 12.5 Gy	600 ng intragastrically on days -1, 1, 3 and 5	3 wk	↓ inflammatory cells infiltration in BALF, ↓ total protein in BALF, ↓ histopathological changes in the lungs	↓ IL-6, ↓ TGF-β, ↑ IL-10, ↓ fibronectin, ↓ α-SMA, ↓ collagen I	[173]
		Radiation-induced pulmonary fibrosis in C57BL/6 mice was induced by thoracic irradiation with a single dose of 22.5 Gy	600 ng intragastrically on days -1, 1, 3, 5, 7 and 9	12 wk	↓ collagen deposition	↓ fibronectin, ↓ α-SMA, ↓ collagen I	
Friedelane	Celastrol	A549 treated with TGF-β1 (5 ng/mL)	1	30 min with celastrol, then 24 h–72 h with TGF-β1	↓ morphological changes (72 h), ↓ invasion (36 h)	↑ E-cadherin (72 h), ↓ Snail (24 h)	[152]
		A549 treated with TGF-β1 (5 ng/mL)	5	24 h		↑ E-cadherin, ↑ ZO-1, ↓ N-cadherin, ↓ vimentin, ↓ Snail, ↓ Slug	[174]
		Pulmonary fibrosis in Wistar albino rats was induced by single intratracheal administration of bleomycin (3 U/kg)	5 mg/kg, twice a week	28 d		↓ TGF-β1, ↓ p-Smad2/3, ↑ E-cadherin, ↑ claudin, ↓ N-cadherin, ↓ Snail, ↓ Slug, ↓ β-catenin, ↓ Hsp90	
Oleanane	Evoditrilone A	A549	1–2	48 h	↓ colony formation ability ^DD^, ↓ migration ^DD^	↑ E-cadherin ^DD^, ↓ MMP-2 ^DD^, ↓ N-cadherin ^DD^	[148]
Oleanane	Glycyrrhizin	A549 and BEAS-2B treated with TGF-β1 (5 ng/mL)	50–200 (A549); 25–100 (BEAS-2B)	2 h with glycyrrhizin, then 24 h with TGF-β1	↓ migration ^DD^	↓ HMGB1 secretion ^DD^, ↓ HMGB1 ^DD^, ↓ p-Smad2/3 ^DD^, ↑ E-cadherin ^DD^, ↓ vimentin ^DD^	[149]
		A549 and BEAS-2B with lentivirus-mediated HMGB1 overexpression	100 (A549); 50 (BEAS-2B)	24 h		↓ HMGB1 secretion, ↓ HMGB1, ↓ TGF-β1, ↓ p-Smad2/3, ↑ E-cadherin, ↓ vimentin	
Oleanane	Oleanolic acid (OA), OA-loaded P105/TPGS mixed micelles	A549, PC-9	15, 30	24 h	↓ migration (OA-micelles > free OA), ↓ invasion (OA-micelles > free OA)	↑ E-cadherin (OA-micelles > free OA), ↓ N-cadherin (OA-micelles > free OA), ↓ p-ERK (OA-micelles > free OA)	[175]
Friedelane	Pristimerin (Pr)	H1299	0.9–3.6	24–72 h	↓ colony formation ability ^TD^ (1.8 μM Pr), ↓ migration ^TD,DD^, ↓ invasion ^DD^ (48 h)	↓ vimentin (3.6 μM Pr, 48 h), ↓ F-actin (0.9–3.6 μM Pr, 48 h), ↓ integrin β1 (3.6 μM Pr, 48 h), ↓ MMP-2 (0.9–3.6 μM Pr, 48 h), ↓ Snail (0.9–3.6 μM Pr, 48 h)	[151]
Oleanane	Soloxolone methyl	A549 treated with TGF-β1 (50 ng/mL)	0.5	24, 48 h	↓ morphological changes, ↓ migration (24 h, 48 h), ↓ invasion (48 h)	↑ E-cadherin (48 h), ↑ ZO-1 (48 h), ↓ vimentin (48 h), ↓ fibronectin (48 h)	[176]
Oleanane	β-escin (β-Es)	H460	5–40	24 h		↓ ALDH^+^ cell population (5–40 μM β-Es), ↑ p21 (20 μM β-Es; both in ALDH^+^ and ALDH^-^ cells)	[177]
		lung tumors in A/J mice were induced by single intraperitoneal injection of tobacco carcinogen NNK (10 μmol/mouse)	3 weeks after NNK treatment mice were fed with the diet containing 500 ppm β-Es	17, 34 wk	↓ lung tumor formation, ↓ progression of adenomas to adenocarcinomas	↑ p21 (34 wk), ↓ ALDH1A1 (34 wk), ↓ p-Akt (34 wk)	
Oleanane	β-peltoboykinolic acid (β-P)	A549 treated with TGF-β1 (2 ng/mL)	1–10 μg/mL	24–48 h	↓ morphological changes (5 μg/mL β-P, 10 μg/mL β-P; 48 h), ↓ migration ^DD^ (24 h, 36 h)	↑ E-cadherin ^DD^ (48 h), ↓ N-cadherin (10 μg/mL β-P, 48 h), ↓ vimentin ^DD^ (48 h), ↓ fibronectin ^DD^ (48 h), ↓ collagen I ^DD^ (48 h), ↓ p-Smad2 (10 μg/mL β-P, 48 h), ↓ Snail ^DD^ (48 h)	[178]
Lupane	Betulinic acid	293T, A549, H1299	10–30	4 h–7 d	↓ migration (A549, H1299; 20 μM BA > 10 μM BA; 24 h), ↓ invasion (A549, H1299; 20 μM BA > 10 μM BA; 24 h), ↓ the sphere-forming ability (A549, H1299; 20 μM BA > 10 μM BA; 7 d)	the direct binding to Skp2 by forming H-bonds with Lys145, ↓ Skp2-Skp1 interactions (exogenous Flag-Skp1 was transfected in 293T, endogenous Skp2-Skp1 interactions in H1299; 20, 30 μM BA; 4 h), ↓ Skp2-mediated ubiquitination of p27 (exogenous p27 in 293T, endogenous p27 in A549; 10, 20 μM BA; 24 h), ↓ Skp2-mediated ubiquitination of E-cadherin (exogenous E-cadherin in 293T, endogenous E-cadherin in A549; 10, 20 μM BA; 24 h), ↓ Skp2 ^DD,TD^ (A549, H1299), ↑ p27 ^DD,TD^ (A549, H1299), ↑ E-cadherin ^DD,TD^ (A549, H1299)	[179]
		intravenous injection of A549 into BALB/C nude mice (metastasis model)	50 or 75 mg/kg BA was administered on the day 7 after LLC injection	2 mos	↓ metastasis ^DD^		
		A549 and H1299 treated with TGF-β1 (10 ng/mL)	10–20	24 h		↑ E-cadherin ^DD^, ↓ vimentin (A549: 20 μM; H1299 ^DD^), ↓ N-cadherin ^DD^, ↓ Skp2 ^DD^	
		LLC	10–20	24 h	↓ migration ^DD^, ↓ invasion ^DD^	↓ Skp2 ^DD^, ↑ E-cadherin ^DD^	
		subcutaneous injection of LLC into C57BL/6 mice (spontaneous metastasis model)	50 or 75 mg/kg each day after LLC injection	21 d	↓ primary tumor growth ^DD^, ↓ lung metastasis ^DD^	↓ Skp2 ^DD^, ↑ E-cadherin ^DD^, ↑ p27 ^DD^	
		intravenous injection of LLC into C57BL/6 mice (metastasis model)	50 or 75 mg/kg BA was administered on the day 7 after LLC injection	60 d	↓ lung metastasis ^DD^	↓ Skp2, ↑ E-cadherin	
Lupane	Betulinic acid, SYK023	H1299	0.1–30	36 h	↓ migration (BA: 10 μM > 5 μM; SYK023 ^DD^), ↓ invasion (BA: 10 μM > 5 μM; SYK023 ^DD^)	↓ F-actin polymerization (BA: 10 μM > 5 μM; SYK023 ^DD^), ↓ p-FAK (SYK023: 1 μM, 5 μM), ↓ p-Src (BA: 5 μM > 1 μM; SYK023: 0.5–5 μM), ↓ p-Akt (BA: 5 μM; SYK023: 0.5–5 μM), ↓ p-mTOR (SYK023: 1 μM, 5 μM), ↓ N-cadherin (BA: 30 μM; SYK023 ^DD^), ↓ β-catenin (BA: 30 μM; SYK023 ^DD^), ↓ vimentin (SYK023: 20 μM, 30 μM), ↓ c-Myc (BA: 20 μM, 30 μM; SYK023: 20 μM, 30 μM)), ↓ SYPD ^K^ (BA, SYK023: 20 μM)	[180]

* Concentrations are shown in μM unless otherwise stated. ** Time is given in hours (h), days (d), weeks (wk) and months (mos). *** Downward (↓) and upward (↑) arrows indicate suppression and activation of biological process, respectively. **** Downward (↓) and upward (↑) arrows indicate downregulation and upregulation of expression, respectively. ^DD^ Dose-dependent increase in effect. ^TD^ Time-dependent increase in effect. ^K^ Demonstrated as a key mechanism using pharmacological inhibition and/or genetic engineering methods.

## Data Availability

Not applicable.

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
