# Peer review of "The Nexus of Inflammation-Induced Epithelial-Mesenchymal Transition and Lung Cancer Progression: A Roadmap to Pentacyclic Triterpenoid-Based Therapies"

_ijms, 2023, doi:10.3390/ijms242417325_

Round 1

Reviewer 1 Report

Comments and Suggestions for Authors

The review titled "The Nexus of Inflammation-Induced Epithelial-Mesenchymal Transition and Lung Cancer Progression: A Roadmap to Penta-3 cyclic Triterpenoid-Based Therapies" by Odarenko et al. discusses the interplay between inflammatory processes and the exacerbation of malignant characteristics in lung cancer cells, specifically through the mechanism of epithelial-mesenchymal transition (EMT). The review examines the existing literature and provides a roadmap highlighting the crucial role of inflammation in driving EMT and subsequent lung cancer progression. Furthermore, Odarenko et al. propose using triterpenoids (PTs) as a targeted therapeutic approach to disrupt the inflammation-induced EMT process in lung cancer. This suggestion is supported by  evidence demonstrating PTs' anti-inflammatory and anti-cancer properties in various preclinical models.

While the review gives an overview of the topic, some significant concerns need to be addressed.

 ·         The review simplifies the tumor microenvironment by focusing primarily on the effects of inflammation-related markers. However, the tumor microenvironment is highly complex, consisting of various cell types, immune cells, and extracellular matrix components. The interplay between these factors can significantly influence EMT and cancer progression. The authors overlook the complexity of the tumor microenvironment and the potential impact of other components on EMT induction and metastasis. Please add this into discussion. 

·         There is a gap and a lack of clinical evidence. The review primarily refers to studies and experiments conducted in vitro or using animal models. Although these studies provide preliminary data, the lack of clinical evidence limits the direct applicability of the findings to human patients. Further research involving clinical trials is necessary to validate these observations and determine the effectiveness of targeting inflammation-driven EMT in lung cancer treatment. Please add clinical evidence to the discussion. 

·         The review simplifies the complex interactions between various cytokines and signalling pathways involved in inflammation-induced EMT. It mainly focuses on the effects of PTs on the TGF-β signalling pathway, while other crucial pathways influenced by EMT-inducing cytokines are overlooked. A more comprehensive understanding of the molecular mechanisms underlying inflammation-driven EMT is needed to develop effective therapeutic strategies. Please add this to the discussion. 

·         It's not clear why the authors are focused only on PTs. The authors primarily mention PTs as potential inhibitors of inflammation-driven EMT. However, it does not provide a comprehensive analysis or comparison of other phytochemical compounds that may also possess anti-EMT properties. Evaluating a broader range of natural metabolites and their effects on EMT could lead to the identification of more effective therapeutic candidates. The authours should add detailed comparison to the discussion. 

·          The authors mentioned the link between EMT induction and the formation of lung cancer stem cells (CSCs), which highlights the lack of studies on the effect of PTs on CSCs. Considering the role of CSCs in chemoresistance and tumour recurrence, it is essential to investigate the impact of PTs and other compounds on stemness inhibition to fully assess their potential as therapeutic agents.

Some minor notes.

  1. The authors briefly mention the association between severe COVID-19 and EMT but does not elaborate on this relationship or discuss other relevant factors related to COVID-19. Expanding on the impact of COVID-19 on lung cancer and its association with EMT could enhance the relevance and applicability of the review.
  2. Figure 3 needs to be simplified.
  3. The review contains several statements and claims without proper references. It is essential to support the information presented with credible sources to enhance the reliability and validity of the review.

 Overall, the review primarily focuses on summarizing existing knowledge rather than critically analysing the research findings. Including critical insights, discussing the limitations of studies, and addressing conflicting evidence would add depth and credibility to the paper.

Addressing above mentioned points would improve the review's clarity, depth, and credibility, making it more informative and valuable for the intended audience. I suggest accepting it for publication after major revision.

Comments on the Quality of English Language

I have no criticisms

Author Response

Dear Reviewer #1,

We are very grateful to you for the time you have taken to analyze our manuscript in detail and for your valuable comments, which have helped to significantly strengthen our review. Let us respond to your remarks.

The review simplifies the tumor microenvironment by focusing primarily on the effects of inflammation-related markers. However, the tumor microenvironment is highly complex, consisting of various cell types, immune cells, and extracellular matrix components. The interplay between these factors can significantly influence EMT and cancer progression. The authors overlook the complexity of the tumor microenvironment and the potential impact of other components on EMT induction and metastasis. Please add this into discussion.

Authors: Corrected. Dear Reviewer #1, as you mentioned, the inflammatory microenvironment consists of cellular and extracellular components. We have described various cell types that induce EMT in lung cancer cells under inflammatory conditions, including CAFs, TAMs, neutrophils, eosinophils, and mast cells (see Sections 2.2.1 and 2.2.2). We have also described the major inflammatory mediators that can induce EMT (see Sections 2.2.1, 2.2.2, and 2.2.3). In a previous version of the manuscript, we underestimated the role of other components of the inflammatory tumor microenvironment, namely the extracellular matrix (ECM) and reactive oxygen species (ROS), in the induction of EMT. Therefore, we have included sections describing these aspects and have shown the ECM and ROS in Figure 2 (see Sections 2.2.4 and 2.2.5 and Figure 2). To facilitate the understanding of the complex relationships between the tumor microenvironment and EMT, we have introduced two new figures summarizing the information from Sections 2.2.1 and 2.2.2 (see p. 5, Figure 3 and p. 7, Figure 4). Thank you for your valuable efforts in deepening our review.

There is a gap and a lack of clinical evidence. The review primarily refers to studies and experiments conducted in vitro or using animal models. Although these studies provide preliminary data, the lack of clinical evidence limits the direct applicability of the findings to human patients. Further research involving clinical trials is necessary to validate these observations and determine the effectiveness of targeting inflammation-driven EMT in lung cancer treatment. Please add clinical evidence to the discussion.

Authors: Corrected. Indeed, clinical trials provide significant evidence for the promise of considering inflammation-induced EMT as a therapeutic target in antitumor therapy, and the lack of description of these data is a serious omission. This section has been added to the manuscript (see Section 4, pp. 13-14, lines 554-609).

The review simplifies the complex interactions between various cytokines and signalling pathways involved in inflammation-induced EMT. It mainly focuses on the effects of PTs on the TGF-β signalling pathway, while other crucial pathways influenced by EMT-inducing cytokines are overlooked. A more comprehensive understanding of the molecular mechanisms underlying inflammation-driven EMT is needed to develop effective therapeutic strategies. Please add this to the discussion.

Authors: Indeed, the literature analysis revealed that the vast majority of publications devoted to the study of the anti-EMT activity of PTs in lung cancer cells focused only on the analysis of the TGF-β signaling pathway, while other signaling axes and regulators are practically not investigated in this topic. This fact is indicative of the very early stage of investigation of PTs as potential blockers of inflammation-induced EMT. One of the goals of this review was to highlight the gaps in our understanding of the mechanisms of anti-EMT activity of PTs, which can be used as a roadmap for further research. A sentence describing this aspect is presented in Section 6, p.27, lines 140-144.

It's not clear why the authors are focused only on PTs. The authors primarily mention PTs as potential inhibitors of inflammation-driven EMT. However, it does not provide a comprehensive analysis or comparison of other phytochemical compounds that may also possess anti-EMT properties. Evaluating a broader range of natural metabolites and their effects on EMT could lead to the identification of more effective therapeutic candidates. The authours should add detailed comparison to the discussion.

Authors: Corrected. In the present review, we wanted to focus the reader's attention only on PTs without distracting the reader from other classes of natural metabolites, reviews on the anti-EMT activity of which are widely available in PubMed. However, it would indeed be remiss not to mention these compounds in our manuscript. Therefore, we have listed a number of key phytochemicals with proven anti-EMT potency and provided references to recent comprehensive reviews on this topic in Section 4 (please see p.15, lines 611-617).

The authors mentioned the link between EMT induction and the formation of lung cancer stem cells (CSCs), which highlights the lack of studies on the effect of PTs on CSCs. Considering the role of CSCs in chemoresistance and tumour recurrence, it is essential to investigate the impact of PTs and other compounds on stemness inhibition to fully assess their potential as therapeutic agents.

Authors: Corrected. Dear Reviewer 1, you rightly point out the need to pay special attention to the effect of PTs on stemness inhibition, but as mentioned above, PTs have been poorly studied in the context of CSCs associated with inflammation-driven EMT. To make the lack of information in this area more explicit, a number of sentences on this topic have been introduced in Section 6 (see p. 27, lines 156-164).

The authors briefly mention the association between severe COVID-19 and EMT but does not elaborate on this relationship or discuss other relevant factors related to COVID-19. Expanding on the impact of COVID-19 on lung cancer and its association with EMT could enhance the relevance and applicability of the review.

Authors: Corrected. Indeed, the relationship of COVID-19 with lung cancer progression is of great interest to researchers, so supplementing this review with information on the EMT-stimulating effect of SARS-CoV-2 in lung tissue seems highly advisable. Thank you for this comment. Section 2.2.6 entitled "COVID-19-associated inducers of EMT in lung tissue" has been added to the manuscript (see pp. 11-12, lines 448-473).

Figure 3 needs to be simplified

Authors: Corrected. The illustration, called Figure 3 in the old version and Figure 6 in the new revised version of the manuscript, has been simplified in several ways (see p. 17, Figure 6):

(1) We have reduced the detail of ubiquitin-mediated degradation by simply showing an inhibition arrow labeled "+Ub" from the ubiquitin ligases Skp2-SCF and Smurf1 to their targets E-cadherin and Rho, respectively.

(2) We masked RAGE and left TLR4 as a putative receptor for HMGB1 (the exact mechanism of how glycyrrhizin inhibits HMGB1-induced EMT is unclear).

(3) We have reduced the number of coregulators and adaptor proteins, leaving only the key participants in the pathways described.

(4) We omitted the involvement of integrins in TGF-β maturation because it is not relevant for triterpenoid activity in lung cancer cells.

(5) We have transferred the structures of pentacyclic triterpenes to a figure describing the pathways of their biosynthesis, as suggested by Reviewer #2 (see p. 16, Figure 5). Structures from Figure 7 have been removed in the same way (see p. 26, Figure 7).

The review contains several statements and claims without proper references. It is essential to support the information presented with credible sources to enhance the reliability and validity of the review.

Authors: Corrected. Please see p. 2 (lines 83, 86, 90), p. 5 (line 151), p. 6 (lines 173, 187, 203), p. 7 (line 230), p. 8 (lines 250, 257, 270), p. 9 (lines 302, 309, 320, 334), p. 24 (line 50).

Overall, the review primarily focuses on summarizing existing knowledge rather than critically analysing the research findings. Including critical insights, discussing the limitations of studies, and addressing conflicting evidence would add depth and credibility to the paper

Authors: Corrected. In fact, we have already provided critical analysis and identification of limitations of the reviewed studies in the Conclusions section of the previous version of the manuscript. In order to make these paragraphs more visible to the reader, we have moved them to a separate section 6 "Future perspectives and limitations" and made some adjustments (see pp. 26-27, lines 125-167).

We hope that corrected version of the manuscript will be acceptable for publication in the International Journal of Molecular Sciences.

Sincerely,

Dr. Andrey Markov

Reviewer 2 Report

Comments and Suggestions for Authors

The manuscript entitled "The Nexus of Inflammation-Induced Epithelial-Mesenchymal Transition and Lung Cancer Progression: A Roadmap to Pentacyclic Triterpenoid-Based Therapies," authored by Kirill V. Odarenko, Marina A. Zenkova, and Andrey V. Markov, analyzes recent studies and emphasizes the role of epithelial-mesenchymal transition (EMT) as a link between inflammation and lung cancer. In the inflammatory tumor microenvironment (iTME), fibroblasts, macrophages, granulocytes, and lymphocytes produce inflammatory mediators, some of which can induce EMT.

The manuscript is clear in all its parts, providing information that leads to increased invasiveness of tumor cells and self-renewal of cancer stem cells (CSCs), associated with metastasis and tumor recurrence, respectively. Consequently, I believe that it can indeed contribute to the current state of the art, and it would be a shame not to consider it for publication in IJMS.

The review is well-written, with illustrative images explaining the presented content. Overall, the manuscript is precise in conveying information and reporting content of high scientific value. However, before deeming it suitable for publication, some minor modifications are necessary:

To enhance discoverability on prominent scientific search engines such as PubMed, Google Scholar, and Scopus, it is recommended to strategically refine selected keywords. This involves eliminating redundant terms while introducing new, relevant ones. The primary purpose of these keywords is to facilitate accurate indexing, as search engines heavily rely on the terminology found in the article's title, abstract, and keywords. Using keywords that duplicate existing content within these sections could diminish the article's visibility. Therefore, a judicious approach includes replacing repetitive keywords with fresh alternatives to augment the article's potential impact upon resubmission.

While the introduction is well-written, specific information regarding pentacyclic triterpenoids is lacking. I understand that the article's focus is on inflammation, but it is essential to introduce additional information regarding the potential biological activities of this class of bioactive compounds, even beyond their anti-inflammatory activity. For example: (i) Ursanes, oleananes, and tirucallanes from Protium heptaphyllum have recently been linked to the inhibition of cholesterol synthesis and release, making them suitable candidates as hypocholesterolemic agents (doi: 10.3390/ijms22052664); (ii) Boswellic acids from Boswellia serrata showed ameliorative effects related to antihyperglycemic, antihyperlipidemic, and wound healing in diabetic rats (doi: 10.21929/abavet2020.35); (iii) Ursolic acid is also implicated in the treatment of anxiety disorders (doi: 10.1016/j.ejphar.2015.03.077); (iv) Oleanolic acid was found to suppress mustard oil-induced nociceptive behaviors (doi: 10.1248/bpb.29.82). I strongly suggest incorporating this information along with the respective bibliographic references.

The manuscript is a review on the potential anti-inflammatory actions of pentacyclic triterpenes of plant origin. I expected to find at least one section where greater attention is given to the class of molecules rather than exclusively focusing on their actions. As mentioned earlier, some information should be introduced in the introduction, providing a general overview of other biological actions outside the scope of this article. However, before delving into cellular mechanisms and potential anti-inflammatory mechanisms, it would be appropriate to introduce a brief paragraph regarding the distribution of pentacyclic triterpenes in plant matrices, along with a description of their chemistry and possibly the biosynthetic pathway. Some information can be found in the following articles: (i) 10.1039/D2NP00063F; (ii) 10.3390/molecules14062016; (iii) 10.1021/jf102039t.

The concluding section is excessively lengthy and should be drastically reduced to two or three paragraphs at most. Additional information presented in this section could either be integrated into earlier sections or eliminated if repetitive.

Author Response

Dear Reviewer #2,

Thank you for taking the time to read and thoroughly analyze our article. We revised the manuscript according to your comments, and, please, let us respond to them.

To enhance discoverability on prominent scientific search engines such as PubMed, Google Scholar, and Scopus, it is recommended to strategically refine selected keywords. This involves eliminating redundant terms while introducing new, relevant ones. The primary purpose of these keywords is to facilitate accurate indexing, as search engines heavily rely on the terminology found in the article's title, abstract, and keywords. Using keywords that duplicate existing content within these sections could diminish the article's visibility. Therefore, a judicious approach includes replacing repetitive keywords with fresh alternatives to augment the article's potential impact upon resubmission.

Authors: Corrected. We are very grateful for this valuable comment. Previously, we approached the writing of keywords only from a formal point of view and thus lost the visibility of our works. Now we will pay more attention to this issue in our next publications. The corrected list of keywords can be found on p. 1, lines 26-27.

While the introduction is well-written, specific information regarding pentacyclic triterpenoids is lacking. I understand that the article's focus is on inflammation, but it is essential to introduce additional information regarding the potential biological activities of this class of bioactive compounds, even beyond their anti-inflammatory activity. For example: (i) Ursanes, oleananes, and tirucallanes from Protium heptaphyllum have recently been linked to the inhibition of cholesterol synthesis and release, making them suitable candidates as hypocholesterolemic agents (doi: 10.3390/ijms22052664); (ii) Boswellic acids from Boswellia serrata showed ameliorative effects related to antihyperglycemic, antihyperlipidemic, and wound healing in diabetic rats (doi: 10.21929/abavet2020.35); (iii) Ursolic acid is also implicated in the treatment of anxiety disorders (doi: 10.1016/j.ejphar.2015.03.077); (iv) Oleanolic acid was found to suppress mustard oil-induced nociceptive behaviors (doi: 10.1248/bpb.29.82). I strongly suggest incorporating this information along with the respective bibliographic references.

Authors: Corrected. We agree that the diversity of bioactivities of pentacyclic triterpenoids should be mentioned for better characterization. Thank you for pointing out some studies on this topic, we have included this information in the revised manuscript (please see p. 15, lines 640-646).

The manuscript is a review on the potential anti-inflammatory actions of pentacyclic triterpenes of plant origin. I expected to find at least one section where greater attention is given to the class of molecules rather than exclusively focusing on their actions. As mentioned earlier, some information should be introduced in the introduction, providing a general overview of other biological actions outside the scope of this article. However, before delving into cellular mechanisms and potential anti-inflammatory mechanisms, it would be appropriate to introduce a brief paragraph regarding the distribution of pentacyclic triterpenes in plant matrices, along with a description of their chemistry and possibly the biosynthetic pathway. Some information can be found in the following articles: (i) 10.1039/D2NP00063F; (ii) 10.3390/molecules14062016; (iii) 10.1021/jf102039t.

Authors: Corrected. Dear Reviewer #2, thank you for pointing out this omission in our article. We have included the information on the biosynthetic pathways of pentacyclic triterpenoids and provided a graphical representation of them (Please, see p. 15, lines 620-628, Figure 5). Representative plant sources of the four main types of pentacyclic triterpenoids were mentioned in the same section (see p. 15, lines 629-639). Although we have not discussed these issues in detail, we have provided references to studies that describe them in more detail, including those that you provided to us.

The concluding section is excessively lengthy and should be drastically reduced to two or three paragraphs at most. Additional information presented in this section could either be integrated into earlier sections or eliminated if repetitive.

Authors: Corrected. The concluding section from the previous version of manuscript was divided into two chapters, including Section 6 "Future prospects and limitations" and Section 7 "Conclusions" (please see pp. 26-27, lines 125-178). Thus, the concluding section has been drastically reduced in accordance with your comment.

We hope that this version of the manuscript will be acceptable for publication.

Thank you very much!

Sincerely,

Dr. Andrey Markov

Round 2

Reviewer 1 Report

Comments and Suggestions for Authors

The authours have successfully addressed all my comments.